# Commensal gut bacteria employ de-chelatase HmuS to harvest iron from heme

Arnab Kumar Nath [1,3], Ronivaldo Rodrigues da Silva [1,3], Colin C Gauvin [1,3], Emmanuel Akpoto [1], Mensur Dlakić [2], C Martin Lawrence [1✉] & Jennifer L DuBois [1✉]

## Abstract

Iron is essential for almost all organisms, which have evolved different strategies for ensuring a sufficient supply from their environment and using it in different forms, including heme. The *hmu* operon, primarily found in Bacteroidota and ubiquitous in gastrointestinal tract metagenomes of healthy humans, encodes proteins involved in heme acquisition. Here, we provide direct physiological, biochemical, and structural evidence for the anaerobic removal of iron from heme by HmuS, a membrane-bound, NADH-dependent de-chelatase that deconstructs heme to protoporphyrin IX (PPIX) and Fe(II). Heme can serve as the sole iron source for the model gastrointestinal bacterium *Bacteroidetes thetaiotaomicron*, when active HmuS is present. Heterologously expressed HmuS was isolated with bound heme molecules under saturating conditions. Its cryo-EM structure at 2.6 Å resolution revealed binding of heme and a pair of cations at distant sites. These sites are conserved across the HmuS family and chelatase superfamily, respectively. The proposed structure-based mechanism for iron removal by HmuS is chemically analogous to the chelatases in both unrelated heme biosynthetic pathways and homologous enzymes in the biosynthetic pathways for chlorophyll and vitamin B12, although the reaction proceeds in the opposite direction. Taken together, our study identifies a widespread mechanism via which anaerobic bacteria can extract nutritional iron from heme.

**Keywords** Heme; Iron; Microbiome; Chelatase; Porphyrin
**Subject Categories** Metabolism; Microbiology, Virology & Host Pathogen Interaction

## Introduction

Iron is the flint that supplies the catalytic spark for nearly all cellular life, and a growth-limiting nutrient for humans and their resident microbiomes. Aerobic life forms, including pathogens in the highly aerated niches of the blood, lung, and urinary tract, depend on iron for trafficking, sensing, activating, detoxifying, and deriving respiratory energy from $O_2$. By contrast, the parts of the gastrointestinal (GI) tract where dietary iron absorption occurs are anaerobic at their core and dominated by anaerobes from the Bacteroidetes and Firmicutes phyla (Shin et al, 2024; Ijssennagger et al, 2015). These species require iron for harvesting energy and for building biomass from the host's diet. At the same time, the microbiota may facilitate host iron uptake and modulate the risk of diseases related to iron mismanagement, particularly anemia, inflammatory diseases, and cancers (Ijssennagger et al, 2015). In addition, pathological niches outside the GI tract, including tumors, anoxic and necrotic wounds, atherosclerotic plaques, and the subgingival layer, harbor pathogenic anaerobes from the same genera (Metwaly et al, 2022; Constante et al, 2017).

Most of the 2–3 g of iron in the healthy human body is heme-associated, and new hemoglobin-packed red blood cells are produced at a rate of 2.3 million per second (Muckenthaler et al, 2017; Pasricha et al, 2021). To meet these high metabolic demands, damaged red blood cells are assiduously recycled by macrophages, in which the heme oxygenase/biliverdin reductase system converts heme into iron, carbon monoxide, and bilirubin. The iron is salvaged, and bilirubin is sent first to the liver and then the intestines for further metabolism (White et al, 2013). Dietary heme and heme derived from continuous recycling of the intestinal epithelium, by contrast, bypasses this heme reclamation cycle altogether, entering the gastrointestinal tract (GI) directly where it is completely metabolized. How this happens is unclear. On the host side, a transport protein for direct uptake of heme into nutrient-absorbing, human intestinal epithelial cells (enterocytes) has not been unequivocally identified (Shayeghi et al, 2005; Dutt et al, 2022; Andrews, 2007). Among microbiome species, Bacteroidetes that dominate the GI tract must import heme, which they absolutely require but cannot synthesize (Gruss et al, 2012). Reflecting their complex iron needs, we and others have observed that members of the Bacteroides genus and many in the Bacteroidetes phylum are highly susceptible to iron deprivation. They are rapidly and selectively diminished or eliminated by an iron challenge in mouse models (Coe et al, 2021).

We recently showed that *Bacteroides thetaiotaomicron* VPI 5482 (*B. theta*), an obligate anaerobe, heme auxotroph, and a model GI tract species (Trosvik and de Muinck, 2015), can use heme as its

[1]Department of Chemistry and Biochemistry, Montana State University, Bozeman, MT 59717, USA. [2]Department of Microbiology and Cell Biology, Montana State University, Bozeman, MT 59717, USA. [3]These authors contributed equally: Arnab Kumar Nath, Ronivaldo Rodrigues da Silva, Colin C Gauvin. ✉E-mail: c.martin.lawrence@gmail.com; jennifer.dubois1@montana.edu

only source of iron under anaerobic conditions (Meslé et al, 2023). This result is remarkable, since well-described heme disassembly mechanisms use $O_2$ to cleave the PPIX macrocycle, freeing the iron (Wilks and Ikeda-Saito, 2014; Murdoch and Skaar, 2022). An anaerobic, radical S-adenosyl methionine-dependent, heme-cleaving pathway encoded by the *chu* operon was recently identified in a handful of pathogens (Brimberry et al, 2023; Mathew et al, 2022; LaMattina et al, 2016; McGregor et al, 2023). A more widespread role for this pathway among the commensal anaerobes of the GI tract has not been demonstrated. *B. fragilis* and *Porphyromonas gingivalis*, both pathogenic anaerobes, have recently been proposed to break down heme by separating iron from PPIX in a de-chelation reaction, leaving the porphyrin intact (Smalley and Olczak, 2017; Rocha et al, 2019). Biological de-chelation reactions, however, have never been directly demonstrated in any species, and their existence remains controversial (Rocha et al, 2019; Létoffé et al, 2009; Dailey et al, 2011).

Here, we provide a comprehensive genetic, physiological, structural, and sequence-based description of the enzyme encoded by *hmuS*, part of a 6-gene operon for heme utilization (*hmuYRSTUV*) that pervades members of phylum Bacteroidetes

and the metagenomes of healthy human GI tracts (Fig. 1A) (Meslé et al, 2023), definitively identifying it as a functional heme de-chelatase. Phenotypic analyses of *B. theta* and its transposon insertion mutants (Arjes et al, 2022) demonstrate that HmuS, a large (1469 residues, 163 kDa), multidomain membrane protein (Fig. 1B), is required for using heme as an iron source. Functional studies of the heterologously expressed HmuS protein demonstrate that it binds multiple heme molecules, anaerobically converting heme to PPIX and Fe(II)$_{aq}$ in an NADH-dependent manner. We report the first three-dimensional structure for a protein from this family in its heme- and cation-bound form, obtained at 2.6 Å resolution by cryo-electron microscopy (cryoEM). We enumerate key differences between HmuS and homologous (class 1) chelatases responsible for an analogous reverse reaction: insertion of divalent manganese and cobalt, respectively, into chlorophyll (ChlH) and vitamin B12 (CobN) in their aerobic syntheses (Romão et al, 2011; Zhang et al, 2021; Lundqvist et al, 2009; Adams et al, 2020; Chen et al, 2015; Wu et al, 2022). The widespread bacterial heme lyase activity encoded by HmuS may be important for assimilation of heme iron by both the microbiome and human enterocytes, playing critical

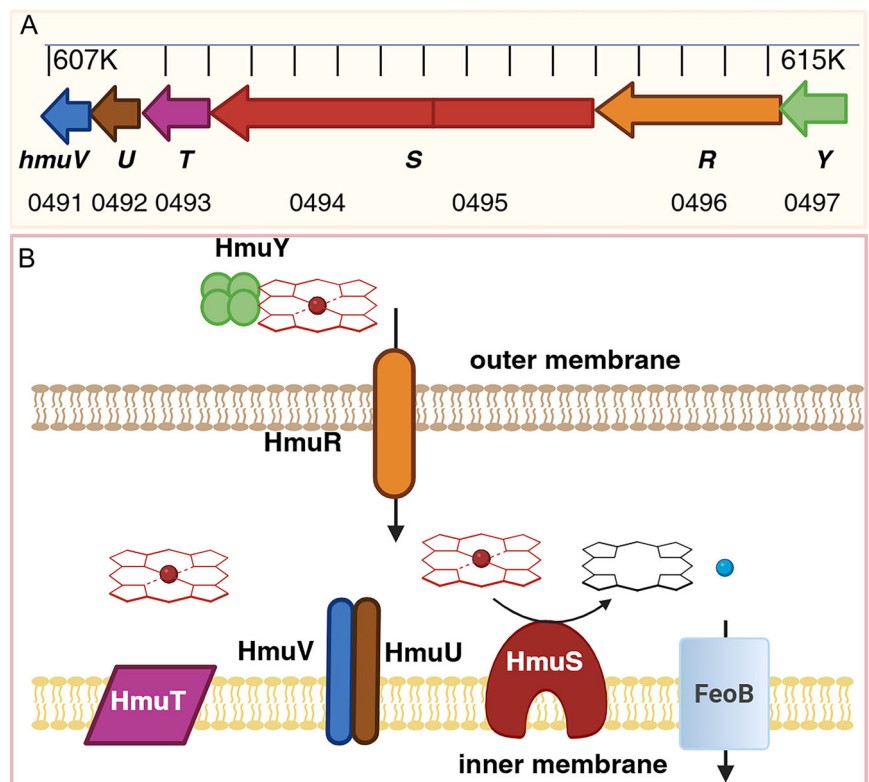

**Figure 1. Organization of genes and proteins encoded by the heme utilization operon from *Bacteroides thetaiotaomicron* VPI-5482.**

(A) Magnified region of the complete genome between nucleotides 607K–616K (GenBank accession AE015928.1). Arrows indicate each of the *hmuYRSTUV* genes encoded on the antisense strand. BT loci numbers are given beneath the gene names. The *hmuS* gene was erroneously frame-shifted and assigned to two loci in the initial genome assembly, though the error was corrected subsequently (GenBank assembly GCA_900624795.1). (B) Predicted cellular organization of gene products. HmuY, a hemophore, is a monomeric lipoprotein that forms a soluble, heme-binding tetramer (Wójtowicz et al, 2009). HmuR is a TonB-dependent outer membrane heme transporter that, in conjunction with HmuV and HmuU, uses the inner membrane electrochemical gradient to power heme import to the periplasm. HmuS removes iron from heme, generating PPIX. HmuT is a predicted inner membrane protein that may form part of a larger complex involved in inner membrane heme transport. Its function is unclear. FeoB is a divalent metal transporter, not part of the *hmu* operon, that imports Fe$^{2+}$(aq) (blue sphere) into the cytoplasm. Heme and PPIX are diagrammed schematically in red and black sticks, respectively. Source data are available online for this figure.

roles in regulating nutrition, microbiome composition, and iron-associated pathologies.

# Results

## Phenotypic analyses of wild-type (*wt*) and *hmuS* mutant strains show HmuS is required for heme conversion to PPIX and Fe(II)

*B. theta* VPI-5482 is a model GI-tract species encoding just one copy of the intact *hmu* operon. The operon, in turn, contains the strain's single annotated *cobN* homolog, distinguishing it from a previously characterized strain of virulent *B. fragilis* and simplifying its genetic characterization (Rocha et al, 2019). Cultures of *B. theta* were grown on a minimal medium including trace non-heme iron ($\leq 1\,\mu M$) and added hemin (15 $\mu M$ ferric protoporphyrin IX chloride) as iron sources. Cultures were monitored optically over time until saturation (Fig. 2A, purple lines), then pelleted and lysed in a solvent that co-extracts heme and PPIX with equal efficiency (Appendix Fig. S1) (da Silva et al, 2024). Extracts made from equivalent cell masses were analyzed for heme and PPIX by high-performance liquid chromatography (HPLC), using their previously verified retention times and mass spectra (da Silva et al, 2024) (Appendix Figs. S1 and S2). These cells accumulated substantial heme ($40 \pm 0.9\,\mu M$) relative to a much smaller quantity of PPIX ($0.035 \pm 0.006\,\mu M$) (Fig. 2B; Appendix Fig. S3).

The experiment was repeated but with increasing concentrations (50–300 $\mu M$) of bathophenanthroline disulfonic acid (BPS), a non-metabolizable chelator that sequesters non-heme $Fe^{2+}$, added to the growth medium. These cells experienced a longer lag phase before entering exponential growth and saturated at lower optical density (Fig. 2A, salmon and red lines). Moreover, while the heme concentration in cell pellets was similar to the unstressed condition, PPIX increased by more than an order of magnitude (Fig. 2B; Appendix Fig. S3). *B. theta* is incapable of making PPIX biosynthetically, and control experiments verified that PPIX is not generated from hemin in the absence of cells. These results therefore indicate that *B. theta* can both incorporate heme into the cell mass and catabolize it to meet its non-heme iron needs (for example, supplying iron for FeS clusters), leaving PPIX as a byproduct. Heme usage is stimulated by restricting the available $Fe^{2+}(aq)$.

We next grew two strains of *B. theta* in which *hmuS* had been inactivated by insertion of a transposon, each at a unique site in the *hmuS* gene (Appendix Fig. S4) (Arjes et al, 2022). Both grew indistinguishably from *wt* in the standard/complete medium (purple lines, Fig. 2C), accumulating similar amounts of heme and a trace amount of PPIX (Fig. 2D). This result suggested that both mutants could meet their cytoplasmic heme and non-heme iron ($Fe^{2+}$) requirements. However, neither strain grew in the heme-sufficient but $Fe^{2+}$-free medium amended with BPS (red lines, Fig. 2C), demonstrating that *B. theta* requires a functional HmuS protein to obtain non-heme iron from heme. As a control, *wt* and mutant *B. theta* strains were also grown with sufficient $Fe^{2+}$ but without added heme. Surprisingly, all grew under these conditions, suggesting that carryover of heme stored by the inoculum could sustain the cultures. When cells were passaged four times into heme-free medium to eliminate heme carryover, however, the mutant strains were viable in a heme/$Fe^{2+}$-sufficient

medium (purple lines) but failed to grow in the presence of $Fe^{2+}$ without added heme (blue lines, Fig. 2D).

## Heme to PPIX conversion depends on the *B. theta* membrane fraction and NADH

The HmuS homologs CobN and ChlH catalyze ATP-driven metal chelation reactions and are encoded alongside genes for ATP-dependent accessory proteins ChlD/ChlI (Walker and Weinstein, 1991; Farmer et al, 2019; Brindley et al, 2015; Adams and Reid, 2013) or CobS/CobT (Lundqvist et al, 2009). We therefore examined the ATP dependence of heme catabolism using lysed *B. theta* cells (0.067 g cell pellet or $1.7 \times 10^9$ *B. theta* colony forming units (cfu) resuspended per mL of 20 mM Tris-HCl, pH 7.1) (Meslé et al, 2023). The whole lysates (300 $\mu L$) were incubated anaerobically with 100 $\mu M$ hemin for 30 or 40 min, after which the cells were extracted to capture PPIX and unreacted heme for HPLC analysis (Appendix Fig. S5). For undialyzed, lysed cells where all of the cytoplasmic small molecules were present, ~85% of the porphyrin content was unaltered heme and 15% was PPIX, with no other porphyrin species detected by UV/visible absorbance spectroscopy (UV/vis) or peaks with different HPLC elution times (Fig. 2E; see Appendix Table S1 for numerical data and errors).

To test whether iron or tetrapyrrole reduction is needed for HmuS-mediated reactivity, assays were carried out under identical conditions but with 1 mM NADH added after extensively dialyzing the cell fractions to remove small molecules. This dramatically reversed the yields of PPIX (71%) and residual heme (29%). ATP had no added effect when co-presented with 1 mM NADH. However, in the presence of 1 mM ATP alone, no residual PPIX was observed. The reasons for the apparent ATP-dependent inhibition of PPIX production are unclear; however, unlike the reactions catalyzed by CobN or ChlH, the heme de-chelation does not require ATP (Adams and Reid, 2013; Debussche et al, 1992) (Fig. 2E; Appendix Fig. S5).

To determine whether catalysis is associated with the membrane fraction (to which HmuS is predicted to be anchored via N- and C-terminal α-helices, Fig. 1B; Appendix Fig. S6), the cell lysate was dialyzed and then fractionated by ultracentrifugation. The supernatant (soluble protein component) and washed/resuspended pellet (membrane protein component) were each anaerobically incubated with 100 $\mu M$ hemin and 1 mM NADH for 40 min before extracting and quantifying PPIX and heme by HPLC. The membrane fraction yielded approximately 80% PPIX and 20% heme after extraction, while the soluble fraction yielded 36% PPIX and 64% heme (Fig. 2F; see Appendix Fig. S5 for representative HPLC data and Appendix Table S1 for numerical data and error bars). These results are consistent with an NADH-dependent, heme-converting activity localized in the membrane fraction, where we hypothesize that the excess PPIX in the soluble fraction could be due to HmuS that is partly solubilized during sonication.

## Heterologous expression and purification yield HmuS in complex with ferrous heme

To examine the de-chelation reaction directly, soluble HmuS was expressed in *E. coli* BL21(DE3)-Lemo cells without its predicted N- and C-terminal membrane-anchoring helices (Appendix Figs. S6 and S7). The HmuS-containing cells produced a pink lysate

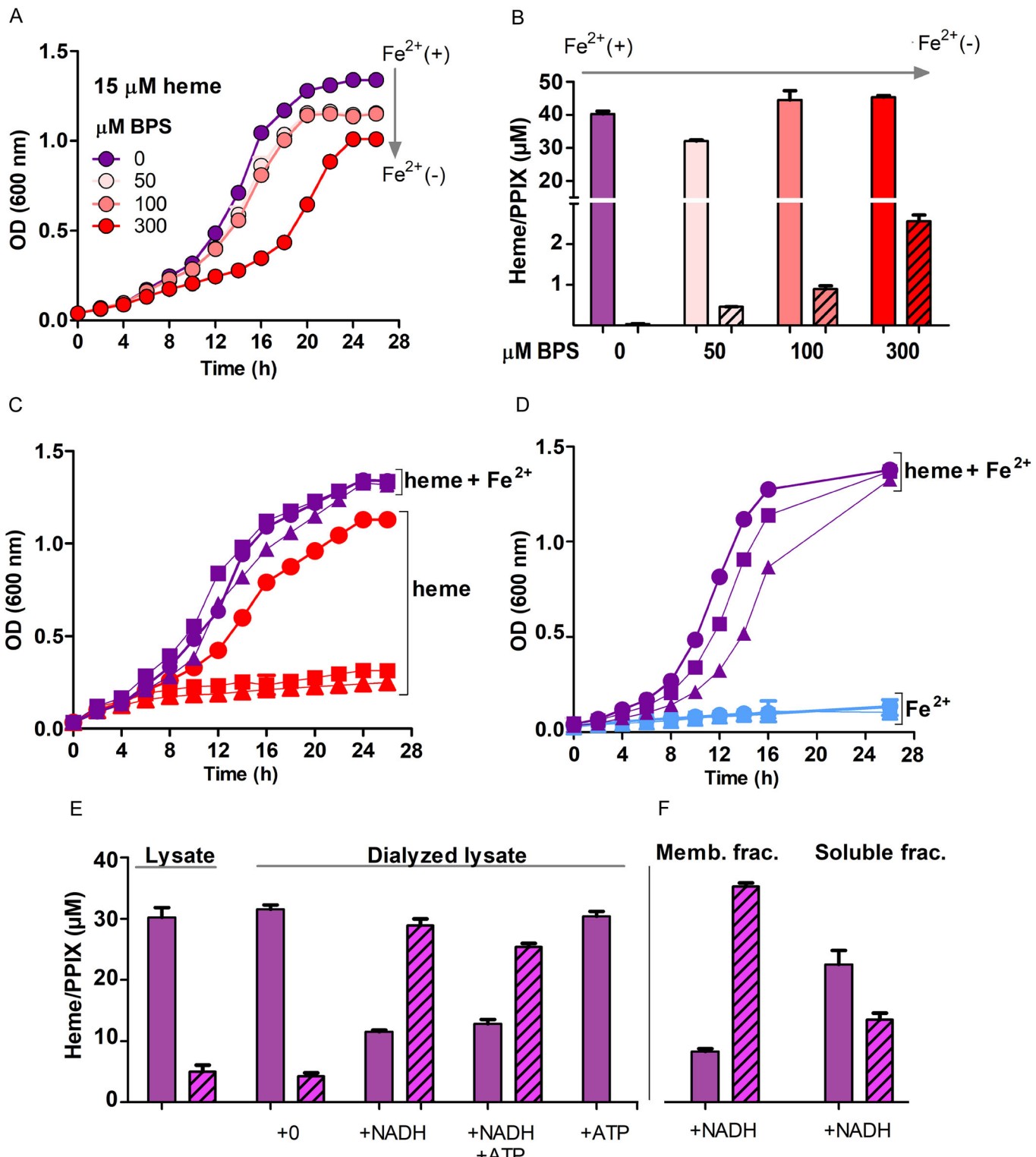

enriched in a protein with the expected HmuS molecular weight (158 kDa) (Appendix Fig. S7). The centrifuge-clarified lysate had a UV/vis absorbance spectrum typical of a 6-coordinate, low-spin ferrous hemoprotein with a prominent Soret band ($\lambda_{max}$ = 425 nm), a weakly absorbing, sharp α band at 560 nm, and a less intense 530 nm β band (Sono et al, 2018) (Fig. 3A). The spectrum

indicating a reduced heme was remarkably stable, remaining unchanged for >24 h in ambient air at room temperature.

HmuS was purified from the clarified lysate using anion exchange chromatography (IEC) in air, 4 °C. The eluted fractions featured a contaminant band near 60 kDa on SDS polyacrylamide gels (SDS-PAGE) that correlated with the density of the HmuS band at 160 kDa

◀ **Figure 2. Heme conversion to PPIX and Fe²⁺ in *B. theta* membranes.**

(A) *B. theta* growth requires non-heme iron. Growth curves illustrate the response of *B. theta* wild-type (*wt*) strain to increasing concentrations of Fe²⁺-sequestering BPS (10 mL minimal medium + 15 µM hemin chloride, inoculated to optical density (OD) at 600 nm = 0.03): 0 µM BPS (purple), 50 µM BPS (light salmon), 100 µM BPS (dark salmon), and 300 µM BPS (red). (B) Depriving cells of Fe²⁺ stimulates metabolism of heme. Heme (solid bar) and PPIX (striped bar) were extracted and quantified from cell pellets collected at the end of the growth experiment. (C) The effects of growing *wt* (●) and *hmuS* mutant (▲, ■) *B. theta* strains on Fe²⁺-free media (red lines) versus Fe²⁺-replete media (purple lines) are shown. When Fe²⁺ is unavailable, the cells with intact *hmuS* metabolize heme for its iron. (D) *B. theta* strains (*wt* (●) and *hmuS* mutant (▲, ■)) were first depleted of stored heme by four serial passages into heme-free media, then inoculated into heme-replete (purple lines) or heme-free media (blue lines). No strains grew when heme was unavailable. (E) The role of added small molecules in the HmuS-catalyzed reaction was monitored. Samples of 300 µL undialyzed/dialyzed *B. theta* cell lysate (1 g cell pellet in 15 mL of 20 mM Tris-HCl buffer, pH 7.1) were incubated anaerobically for 30 min with 100 µM hemin in the presence or absence of 1 mM NADH and/or 1 mM ATP (20 mM Tris buffer, pH 7.1). Reactions were extracted and the product PPIX (striped bars) and unreacted heme (solid bars) quantified by HPLC. (F) The membrane fraction catalyzes heme-PPIX conversion. The dialyzed lysate was fractionated by ultracentrifugation, and the soluble/membrane components incubated for 40 min with 100 µM hemin and 1 mM NADH prior to extraction/analysis, as described in (E). In each case, triplicate results were averaged. Error bars represent standard deviation. Source data are available online for this figure.

(Fig. 3B). Mass spectrometric (MS) analysis of proteins extracted from the 60 kDa and 160 kDa gel bands showed that each was chiefly composed of HmuS peptides (73% and 90%, respectively; Appendix Fig. S7; Dataset EV1). The use of protease inhibitors, metal chelators, centrifuge filtration, and affinity purification (using HmuS with an N-terminal His₆-tag) did not eliminate the 60 kDa band (not shown). We concluded that the large, monomeric HmuS undergoes partial proteolysis during expression and/or purification, and vigilant efforts are required to avoid this.

IEC fractions containing full-length HmuS detected by SDS-PAGE were pooled, yielding an orange-hued solution (Appendix Fig. S8). Further purification by size-exclusion chromatography (SEC) largely removed contaminants below 60 kDa from the red HmuS protein, including a yellow protein that eluted in the 20–30 kDa range (Appendix Fig. S8; see also Dataset EV2 for MS proteomics data for IEC and IEC-SEC purified HmuS fractions and the small molecular weight SEC fractions). After column purification, the UV/vis spectrum for HmuS (Fig. 3A) suggested a ferric, low-spin, 6-coordinate heme, with peak maxima (nm): 415 (Soret), 365 (shoulder), 535 (β), and 565 (α). The narrow, unsplit Soret and pair of α and β bands indicate a single heme-HmuS binding site. Referencing the intensity of the Soret maximum to the concentration of heme released from HmuS in pyridine (pyridine hemochromagen assay) yielded an extinction coefficient for HmuS-heme: $\varepsilon_{(415nm)} \approx 110 \text{ mM}^{-1} \text{ cm}^{-1}$. See Fig. 3B and Appendix Table S2 for typical purification results.

## Formation and reactions of HmuS-heme complexes

Recombinant HmuS is routinely purified with ~10% of the protein bound to the putative substrate, heme. We examined the association between HmuS and heme in several ways. First, we attempted to remove heme from the purified protein by dialysis against 50–300 mM imidazole. However, the protein's UV/vis spectrum and the amount of HmuS-associated heme measured by the pyridine hemochromagen assay were unaffected.

To examine its affinity for heme, we titrated HmuS with hemin, referencing each spectrum to a cuvette to which an equivalent concentration of hemin was added to buffer. The intensities of the Soret, shoulder, and α/β bands (Q-band region) of the HmuS-heme complex already present in the as-isolated protein increased after each incremental addition of heme (Fig. 3C,D). This suggested that the added heme bound the same site as the heme in the as-isolated protein. Fitting a plot of absorbance versus heme concentration

with a quadratic binding curve (Fig. 3C, inset) yielded an apparent $Kd = 6.0 \pm 6$ µM, where the error is the standard deviation from three measurements. Accurate titration of HmuS beyond the highest affinity heme was not possible due to the low solubility and highly chromophoric nature of the heme ligand, and likely overlap in the spectra describing successively bound heme molecules.

To determine whether the high-affinity heme could be reduced, excess NADH (1 mM) was added to HmuS containing substoichiometric heme bound in the highest-affinity site. The ferric HmuS-heme complex converted to a ferrous, low-spin, 6-coordinate complex (Fig. 3E,F) over time, resulting in a UV/vis spectrum closely resembling the clarified *E. coli* lysate (Fig. 3A). No PPIX production was detected by fluorescence spectroscopy. This suggested that reduction of the iron in the high-affinity HmuS-heme complex, on its own, did not convert heme to PPIX and Fe²⁺. Protein-free control reactions containing NADH and hemin likewise did not result in the reduction or demetallation of heme.

Finally, because extended titration was not practical, we sought to populate additional heme-binding sites by equilibrating HmuS with a fivefold excess of hemin. Loosely bound heme was removed using three cycles of centrifuge filtration/resuspension into fresh buffer (Fig. 3G,H). Pyridine hemochromagen analysis indicated that the washed protein (7.0 µM) retained $32 \pm 0.6$ µM (4–5 eq) heme. Following the addition of 350 µM NADH to the multi-heme-loaded HmuS (all reactants freshly prepared and maintained under N₂), a broad NADH-associated peak centered at 340 nm appeared. The spectrum changed slowly over time, stopping within 90 min. The excess unreacted NADH obscured the Soret region of the HmuS-heme spectrum; however, the peaks in the Q-band region converted to a doublet (530, 560 nm), consistent with the spectrum for the reduced HmuS-heme complex (compare Fig. 3H and 3F). The residual heme was measured at the end of the reaction using the pyridine hemochromagen assay. $22 \pm 2$ µM of the initially bound heme remained after the reaction with NADH. These results indicate that HmuS converts heme to PPIX when associated with multiple heme molecules and incubated with excess NADH. However, only 1–2 equivalents of heme were converted to PPIX and Fe²⁺.

## Recombinantly expressed HmuS converts ferroheme to PPIX under catalytic conditions

Finally, we examined recombinant HmuS for heme conversion under conditions similar to those used for *B. theta* cellular fractions. Reactions

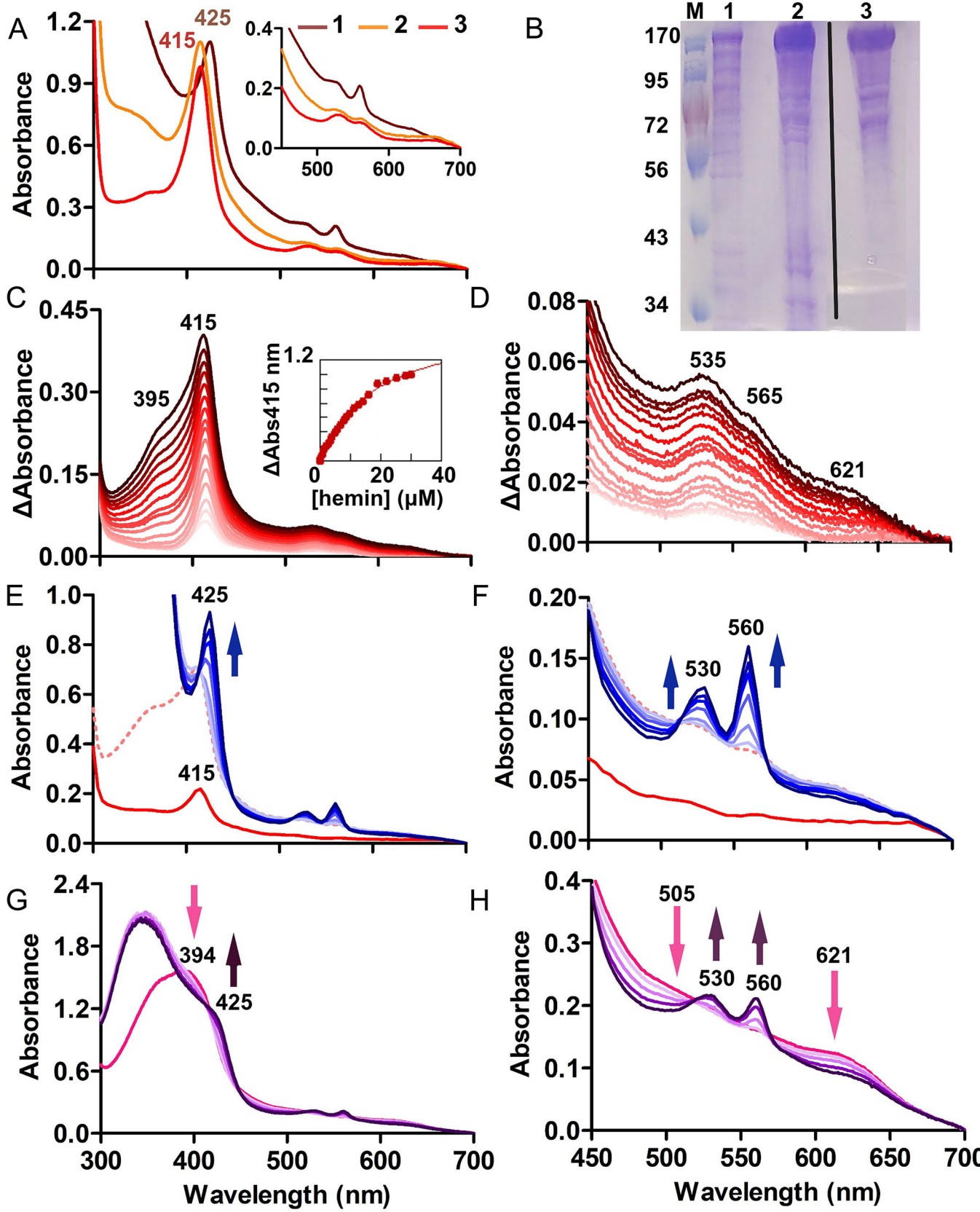

◀

**Figure 3.   Recombinant *B. theta* HmuS, expressed in *E. coli*, binds heme and converts it to PPIX.**

**(A)** Red *E. coli* clarified lysates exhibited a reduced (Fe(II)) heme-protein spectrum following HmuS expression (rust colored line, 1). Following purification by anion exchange (orange line, 2) and size-exclusion (red line, 3) chromatographies, the protein solution became orange and then red. The absorbance spectra indicate an oxidized, 6-coordinate heme protein following column purification. All spectra correspond to 10 μM heme content measured by pyridine hemochromagen assay. **(B)** SDS-PAGE of molecular weight marker (M) and protein fractions from **(A)**. Note that the images showing the marker and purified HmuS are reproduced in Fig. 6B for comparison with mutant protein data. This image is reproduced, in part, in Appendix Fig. S8A. **(C)** HmuS (6 μM, containing 0.6 μM HmuS-heme complex) was titrated in air with a basic hemin solution (pH 8, 0.6 μM aliquots) and monitored by UV/vis absorbance up to 12 μM of hemin added. A cuvette containing buffer alone was titrated in parallel and used as a blank. Difference spectra are shown, with added hemin increasing from light to dark shades of red. Inset: plot of absorbance at the Soret maximum (415 nm) versus [hemin], fit to a quadratic binding equation. **(D)** The Q-band region is shown on an expanded scale. **(E)** HmuS-heme was reduced following the addition of NADH. The as-isolated HmuS protein (20 μM, with ~10% heme bound, solid red line) was incubated with 4 μM ferric heme (dashed red line). Following the addition of 1.0 mM NADH, spectra were measured at 5, 15, 30, 45, 60, and 90 min (blue). The spectral changes are consistent with reduction of heme iron to the ferrous state. Peak positions are labeled. **(F)** The Q-bands are shown on an expanded scale. **(G)** HmuS was incubated with 35 μM heme and buffer exchanged (pink spectrum, 7 μM HmuS with 32 μM bound heme). Spectra were measured every 10 min following the addition of 350 μM NADH (purple lines) under a nitrogen gas atmosphere. The final spectrum (90 min) is shown in dark purple. **(H)** The Q-band region from panel **(G)** is shown on an expanded scale. Source data are available online for this figure.

contained 20 μM HmuS, 200 μM heme, and 1 mM NADH (20 mM Tris pH 7.1, 25 °C, 30 min incubation). PPIX and heme were extracted for analysis by HPLC. $27 \pm 2$ μM PPIX and $140 \pm 3$ μM unreacted heme were detected by HPLC analysis (average of three independent measurements). No product was observed in the absence of NADH, heme, or HmuS (see below). Though HmuS clearly converts heme to PPIX, the reaction is restricted to 1–2 turnovers. Multiple turnovers could be limited by instability of the recombinant protein under the chosen conditions, the need for additional cofactors, product inhibition by hydrophobic PPIX, or other factors. Understanding the limits of the reaction's efficiency will be the subject of future work.

## Structure of HmuS in complex with heme and cations

Cryo-EM single particle analysis was used to determine two related structures of HmuS, each at a global resolution of 2.6 Å (Appendix Figs. S10 and S11). With the exception of a disordered N-terminal domain in the second structure, they are highly similar. Thus, with the exception of the N-terminal domain, the structural description below applies to both structures.

DALI and PDBeFold searches each identified the greatest similarity to cobaltochelatase from *M. tuberculosis* (Mt CobN, PDB ID 7C6O), followed closely by $Mg^{2+}$ chelatase from *Synechocystis* (Syn ChlH, PDB IDs 4ZHJ and 6YSG) (Zhang et al, 2021; Adams et al, 2020; Chen et al, 2015). HmuS de-chelatase shares the overall 6-domain architecture of these class I chelatases, including the N-terminal "head" (domain I) and "neck" (domain II) domains, followed by domains III–VI, which collectively form the body (Fig. 4A). However, structural superposition on CobN or ChlH showed a substantially different conformation of the head domain in HmuS relative to those in both CobN and ChlH, suggestive of a hinge between the neck and body (Fig. EV1). Significantly, HmuS has heme and two cations bound (below), while the other structures are ligand-free.

## Methionine-rich insertion

A structure-based multiple sequence alignment identified a 120-residue, low complexity, methionine-rich insertion (MRI, residues 780–898, 21 methionines) within domain IV (Fig. 4A; Appendix Figs. S13 and S14 highlight secondary and primary structural alignments, respectively). Consistent with low complexity, density for the MRI is absent, suggesting the MRI is intrinsically disordered

in the absence of an appropriate ligand. A conservative set of bona fide HmuS sequences was identified by focusing on complete 6-gene *hmu* operons and selecting only HmuS sequences with at least 50% identity to *B. theta* HmuS. This resulted in 680 non-redundant HmuS sequences that were multiply aligned to identify a set of strictly conserved residues across the HmuS family (Fig. EV2; Appendix Fig. S12; see also Dataset EV3 for tabulated residue conservation scores). The alignment confirmed the presence of an MRI in each HmuS sequence, though they varied in length, showing a mean length of 82 residues, and a minimal length of just 22 (Appendix Fig. S13). The MRI thus appears to be a feature that distinguishes the HmuS de-chelatase family from other members of the class I chelatase superfamily (cobalto- or magnesium chelatases), which lack it (Fig. EV3).

## HmuS body

Like ChlH and CobN, domains III through VI form a globular structure, known as the "body", that houses a central cavity. In the chelatase enzymes, this cavity is proposed to represent the enzyme active site, where the macrocycles are bound and metalated (Adams et al, 2020). The N-terminal half of the body is formed by domains III and IV, in which domain III adopts an α/β fold with a central, predominantly parallel, 6-stranded β sheet ($\beta_{3\uparrow}\beta_{4\uparrow}\beta_{1\uparrow}\beta_{5\uparrow}\beta_{6\uparrow}\beta_{7\uparrow}$), and domain IV is a predominantly helical, extended domain (Fig. 4B,C). Similarly, the C-terminal half of the body is formed by domains V and VI. Domain V is an α/β domain with a central, predominantly parallel, 5-stranded β-sheet ($\beta_{4\uparrow}\beta_{1\uparrow}\beta_{5\uparrow}\beta_{6\uparrow}\beta_{7\downarrow}$), while domain VI is an extended helical structure that contains two successive armadillo-like repeats (α1–α3, α4–α6) followed by a 3-helix bundle (α7–α9). The central cavity lies at the nexus of domains III, V, and VI. When the structure is aligned such that the predicted N-terminal membrane anchor points downward, holding the head domain close to the inner membrane, domain IV runs like a backbone across the top of the structure with major contacts to domains III and VI (Fig. 4C–F).

Mapping the HmuS multiple sequence alignment onto the structure shows the central cavity of HmuS is highly conserved (Fig. 5A–D). Notably, an HFG(T/A)HG motif (residues 534–539) in the β6/α8 loop of domain III and an LSLDHxxEFMGG motif in the β5/α14 loop and α14 of domain V are strictly conserved across the chelatase superfamily and HmuS family, respectively (Appendix Figs. S11–S12; Fig. EV2). His-538 in the first motif and

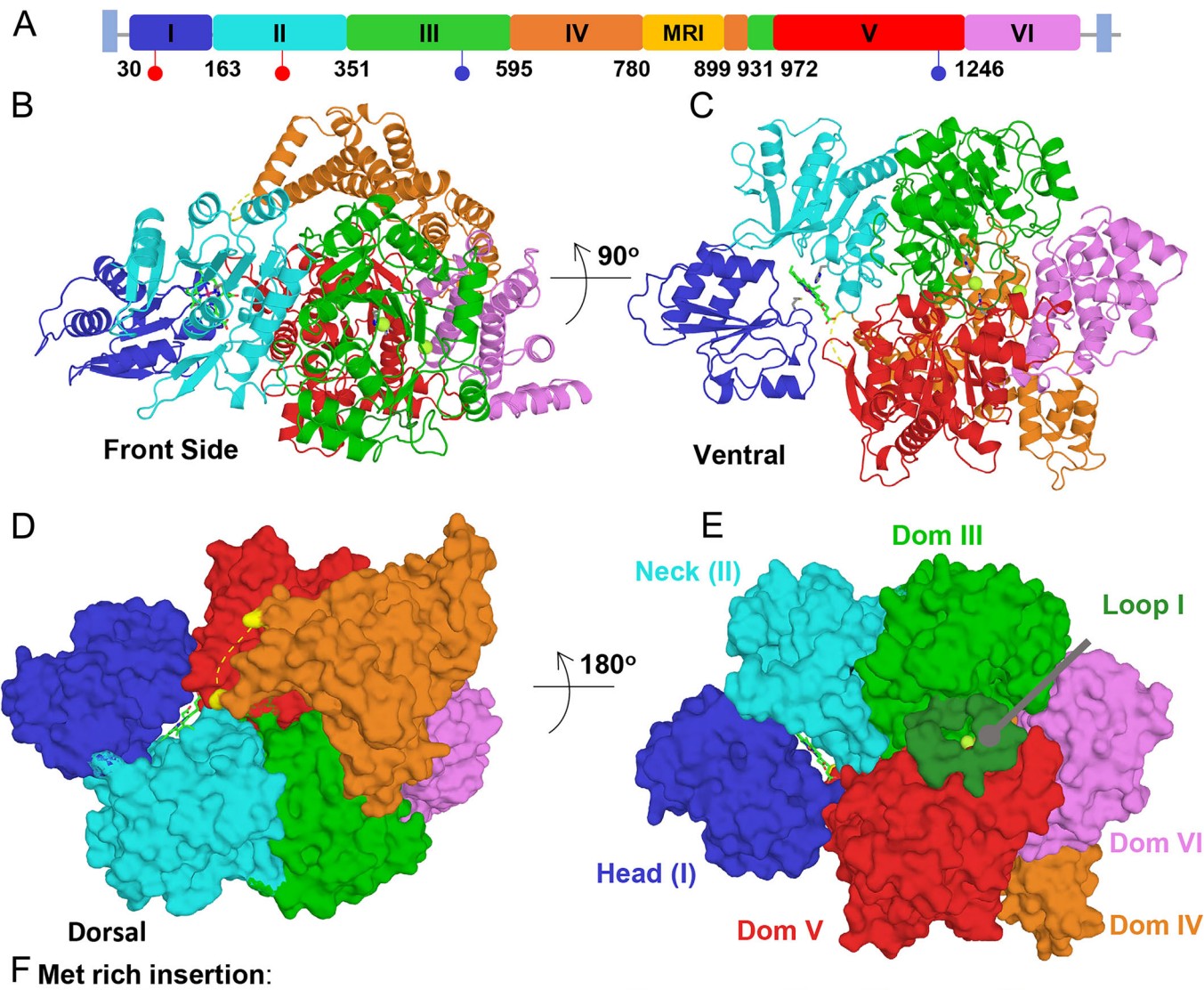

**F** **Met rich insertion**:

GMMAMMMAAAAKKDQADNEPSGNGHPASAKMEKGPHGKMPAGMKEAMKKMGANMDPEKA
MEMAKSMGASPEALKKMEASMKANKDTSTDASGKPAMAGKTEKPQGMSAMMAAMGKAPKE

---

**Figure 4. Structure of HmuS suggests elements of a mechanism by which the >1400 amino acid protein binds and removes iron from heme.**

(A) Domain organization of HmuS. Blue squares at each end indicate transmembrane regions not included in the expression construct. Domains I–VI are named analogously with known class 1 chelatase domains. The first residue number for each domain is shown. Note that domain IV is inserted within domain III. The cryo-EM map lacks density for the 120 residues of the Methionine-Rich Insertion (MRI). Red sticks indicate residues interacting with heme at the head/neck interface (Met-79 and His-254, from left to right), while blue sticks indicate Na1-coordinating residues (His-538 and His-1209, from left to right). (B) Cartoon diagram of HmuS shown from the "front side". Individual domains are colored as in (A). The MRI is not present, but is an insertion in Domain IV (orange), predicted to lie above domains I (dark blue) and II (cyan), which form the N-terminal "head" and "neck" domains of HmuS. The body is formed by domains III–VI, which houses a central cavity formed by domains III, V, and VI, with domain IV running along the top ("backbone"). (C) HmuS is rotated 90 degrees about the horizontal axis, tipping the molecule to expose the membrane-facing underside (ventral view). The heme (green sticks) at the interface of the head and neck domain becomes apparent. (D) Surface representation of (C), ventral view. Domain III is green. Loop I, thought to gate entry into the cleft, is shaded darker forest green. The *bis*-His coordinated sodium ion (Na1) is shown in "limon" and is visible through the Loop I pore. See also Fig. EV4 for views from the reverse side. (E) HmuS from (D) is rotated 180 degrees about the horizontal axis, revealing the "dorsal" view. Residues highlighted in yellow anchor the MRI (dashed lines). (F) MRI sequence with Met residues highlighted in yellow. Source data are available online for this figure.

---

His-1209 in the second extend toward each other, directly across the middle of the putative active site, where they coordinate a strong spherical density, which we have modeled as a sodium cation due to the inclusion of NaCl in the buffer (Figs. 5F, EV4 and EV5). Strictly conserved Glu-1212 forms a hydrogen bond with the His-1209 side chain, likely fixing the position and

fine-tuning the acidity of the imidazole. The observed cation coordination by His-538/His-1209, in conjunction with docking studies, suggests HmuS could utilize these residues for *bis*-His heme coordination within the central cavity. The axial histidines are also well placed to act as general acids, protonating the porphyrin as iron is excised. While strictly conserved in our

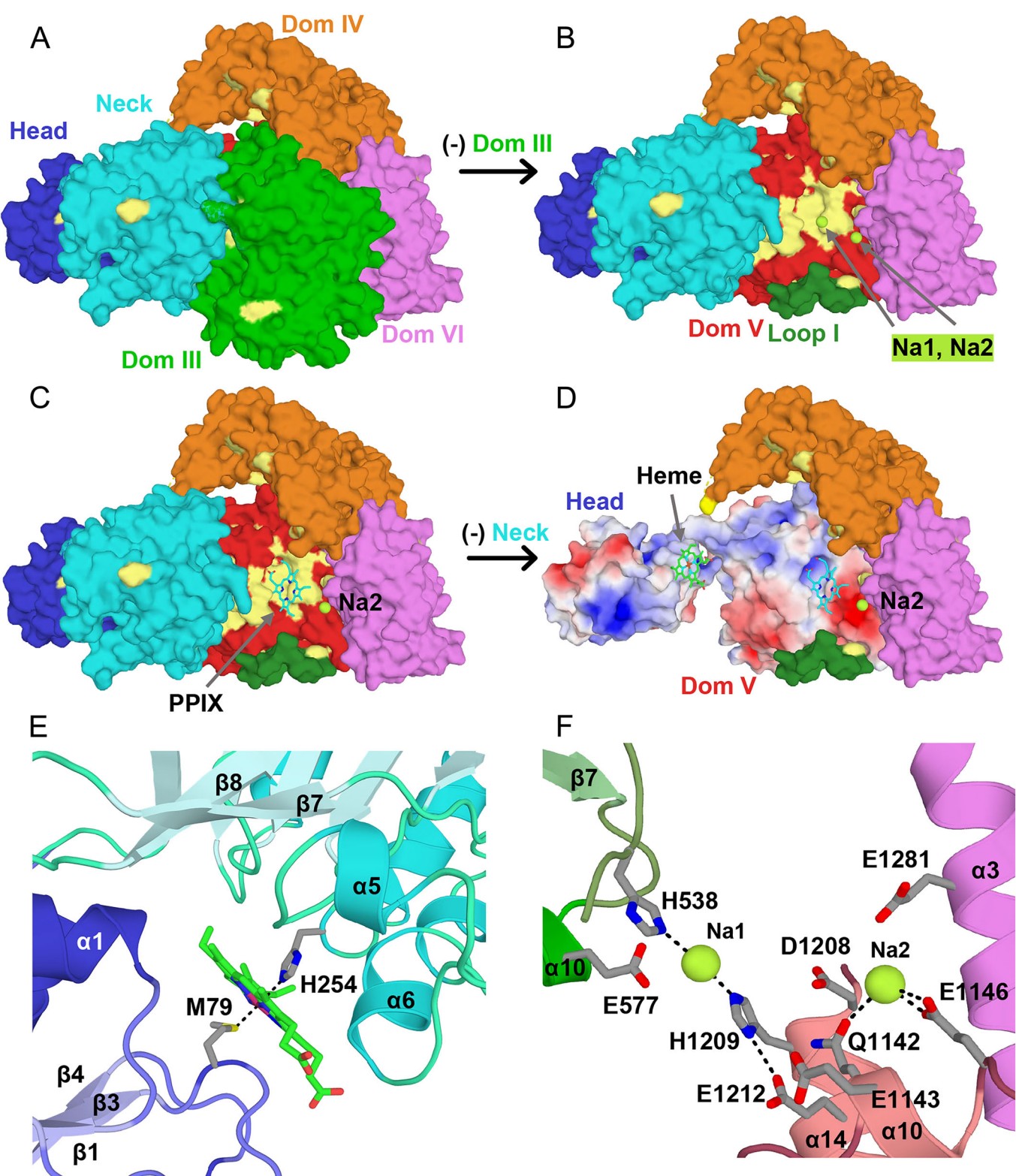

limited set of HmuS sequences, His-538 also appears strictly conserved across the chelatase superfamily. In addition to these motifs, Ser-1014, Ser-1067, Gln-1198, Ser-1199, and Trp1202 are among the strictly conserved residues lining the surface of the HmuS central cleft, and along with the unsaturated N-terminus of

helix α3 in domain V, are predicted to coordinate the propionate groups of a bound heme/PPIX. Consistent with this, while sequence identity to CobN and ChlH are low, structural super-position on CobN or ChlH shows greatest similarity to domains III and V (Dataset EV3).

◀ **Figure 5. Ligand binding sites in HmuS identify functionally significant pockets.**

(A) HmuS is shown in the same "side view" orientation as Fig. 4A, colored by domain. Strictly conserved side chains are in pale yellow; in this view, strict surface conservation is limited. (B) Domain III has been removed from the image, revealing the central cleft with 2 bound Na ions (limon), Na1 and Na2. Loop I, gating access to the central cleft, is also visible in forest green. In contrast to the exterior surface, domain V presents a large, strictly conserved surface within the central cleft (pale yellow on red). Na1 is coordinated by His-538, which lies in the center of this conserved face. (C) The central cleft is the proposed chelatase active site in ChlH and CobN. We therefore docked the HmuS product, PPIX, into the central cleft with AutoDock Vina. The docking model suggests HmuS utilizes the strictly conserved surface on domain V for recognition of PPIX. (D) Domain II (the neck domain, cyan) has also been removed to reveal the heme-binding site at the Head/Neck interface. The heme and PPIX molecules are separated by nearly 30 Å. Electrostatic surfaces are shown for domains I and V (+/− 5kT/e). The propionate groups for both ligands are accommodated by a significant positive charge (blue), and near PPIX they are accompanied by a conserved basic pocket in domain V. (E) Coordination of heme at the interface of the Head and Neck domains is by Met-79 and His-254. The Head domain is in shades of blue, and Neck domain in shades of cyan. In both panels, secondary structure elements are labeled relative to their specific domain. (F) HmuS is shown from the ventral side as in (D). Elements of domains III, V and VI involved in Na$^+$ ion coordination within the central cleft are shown in green, salmon and magenta, respectively. Na1 is coordinated by His-538 and His-1209, with Glu-1212 hydrogen bonding to His-1209. Na2 is coordinated by Gln-1142 and Glu-1146. Residues coordinating Na2 are not strictly conserved. Additional residues contributing to a patch of strong negative charge within the central cleft are also shown (Glu-1143, Asp1208, and Glu-1281). Source data are available online for this figure.

## A non-Pro cis peptide

The potential importance of His-538 may also be indicated by *a cis* peptide bond between Thr-537 and His-538. Interestingly, a *cis* peptide bond is also present in CobN between an equivalent His residue (His-522) and the preceding Lys residue (Zhang et al, 2021). In contrast, a roughly equivalent His residue in ChlH is in the *trans* configuration, with the side chain more distal to the porphyrin metal binding site. Whether these residues undergo *cis/trans* isomerization during the catalytic cycle, as they insert or remove metal ions, remains to be determined.

## Active site accessibility

In ChlH, access to the central cavity is blocked by Loop I (Fig. 4E), which is in a closed conformation (Zhang et al, 2021; Adams et al, 2020; Chen et al, 2015). In contrast, the equivalent loop in CobN was found in an open, partially disordered state (Zhang et al, 2021). Like CobN, the central cavity is also solvent accessible in HmuS, as the β5/α7 loop in domain III is locked in an ordered, "partially open" conformation. However, when modeled in the closed conformation by analogy to ChlH, strictly conserved Phe-502 and His-506 are predicted to lie over the solvent-exposed (propionate distal) edge of the heme, proximal to His-538 and His-1209.

## Structural similarity with class II chelatases

The head (domain I) and neck (domain II) domains show distant structural similarity to those in CobN and ChlH. Domain I exhibits a 4-stranded parallel β-sheet ($\beta_{2\uparrow}\beta_{1\uparrow}\beta_{3\uparrow}\beta_{4\uparrow}$) with right-handed helical crossovers and two additional C-terminal helices, while domain II builds on this core β-sheet structure with two additional strands at the N-terminus to give a 6-stranded β-sheet and a C-terminal β-finger (Figs. 4 and 5). However, unlike CobN and ChlH, the central β-sheets in each of these domains "point" toward each other, with a distinct cleft between the two domains. Given these features, we, and others (Zhang et al, 2021; Adams et al, 2020), note distant similarity between the head and neck domains of the class I chelatase structures with human ferrochelatase (class II), and to the bacterial periplasmic binding protein superfamily in general.

## Heme binding site at the head/neck interface

The HmuS multiple sequence alignment identifies only a few strictly conserved, solvent-exposed residues in the head and neck domains. Notably, these include Met-79 and Gly-80 in the head, as well as a His–Gly–Arg motif (residues 254–256) and Gln-299 in the neck (Figs. 5E and EV4). Each of these residues lies at the interface between the head and neck domains. Critically, we also identified a bound heme at this interface that copurified with HmuS, with the iron coordinated by the strictly conserved Met-79 and His-254 residues (Figs. 5E and EV4). Given the structural similarity of the head and neck domain to class II chelatases and periplasmic binding proteins (above), there has been speculation that class I chelatases might utilize this site to coordinate ligands. However, this is the first time a tetrapyrrole has been found at this site. It should be noted that Met-79 and His-254, while conserved among HmuS sequences, are not universally conserved in the chelatase superfamily, raising the possibility that HmuS binds heme at the head/neck interface in a way that is distinct from how other superfamily members may bind their analogous ligands.

## Disordered head domain in the absence of heme

Superposition of the HmuS head domain on CobN or ChlH requires rotations of 50° and 100°, respectively, away from the neck domain (Fig. EV1). Thus, rather than the "open" conformations seen for CobN and ChlH, the HmuS structure described above captures domain 1 in a "closed" conformation. This raises the question, might HmuS also sample these open conformations?

In this context, density for the head domain in the initial single particle map was relatively well ordered, but significantly weaker than the rest of the structure. This prompted 3D classification efforts focused on domain 1 that resolved two distinct particle sets, one giving a map with strong density for the head domain (PDB 9D26), and a second in which the head domain is disordered (PDB ID 9P4S). Notably, the ordered structure retains strong density for the heme group at the interface of the head and neck domains. In contrast, heme density is absent in the disordered structure. These two structural snapshots are consistent with a mobile head domain in the absence of heme, which adopts an ordered conformation when heme is bound at the subunit interface.

### His-538 is essential for heme turnover to PPIX

Sequence and structural evidence presented above suggest heme binding/demetallation occurs at the Na1 binding site (Fig. 5F). To test this hypothesis, we replaced the strictly conserved His-538 residue with an alanine. The H538A mutant HmuS purified similarly to the wt, with a small amount of heme bound and an identical UV/vis absorbance spectrum (Fig. 6A,B). Further titration of the protein identified a $Kd = 1.4 \pm 0.1\ \mu M$, again similar to wt (Appendix Fig. S14). These results suggest that the protein as a whole and the heme site observed in Fig. 5E are unchanged by this mutation. The H538A mutant protein was then incubated with substrates under conditions that led to heme turnover by the wt protein (Fig. 6C): 20 μM HmuS-H538A, 200 μM heme, 1 mM NADH (20 mM Tris pH 7.1, 25 °C, 30 min incubation). Under these conditions, no PPIX was observed (Fig. 6C). We conclude that the Na1 binding site plays an essential role in the HmuS-mediated reaction.

## Discussion

Well-known cellular mechanisms for releasing iron from heme use $O_2$ to produce a linearized porphyrin, most commonly biliverdin, plus carbon monoxide (CO) (Wilks and Ikeda-Saito, 2014; Wilks and Heinzl, 2014). Some aerobic pathogens undertake similar reactions to yield structurally distinct products (Wilks and Ikeda-Saito, 2014). The portion of the GI-tract in which the microbiome breaks down complex nutrients, however, is anaerobic, as are several disease-associated microenvironments inhabited by human pathogens. Anaerobes from the Bacteroidetes phylum, which require but cannot make heme, are major species in these $O_2$-limited niches (Pasolli et al, 2019), becoming extinct in the GI-tract when dietary iron is unavailable (Coe et al, 2021). A mechanism for liberating heme iron has not been described in commensal anaerobes, making the fate of heme in the GI tract the subject of debate. Here, we used multiple experimental approaches to demonstrate that an anaerobic mechanism, encoded by the *hmuS* gene from the pervasive, Bacteroidetes *hmu* operon (Fig. 1), removes iron from heme, leaving PPIX intact (Coe et al, 2021). This mechanism is sufficiently robust to render heme a sole source of nutritional iron for these microbes.

### Mechanistic insight

Our new understanding of the heme-de-chelation reaction (Figs. 2 and 3), the distribution of *hmu* operons in GI-tract species, and the structure of the HmuS/heme/cation complex (Figs. 4 and 5) opens doors to future research. First, the data presented here allow us to propose a testable model for how tetrapyrrole de-chelation occurs. Solution of the cryoEM structure of HmuS revealed a monomeric 6-domain protein resembling available ligand-free structures of the CobN and ChlH chelatases that have been zoomorphically described as "chicken-like" (Zhang et al, 2021; Adams et al, 2020; Chen et al, 2015). However, unlike these structures, HmuS features a heme molecule tucked between the craning head and neck (domains I–II) (Figs. 4 and 5). The heme iron was axially ligated by Met-79 and His-254, residues conserved

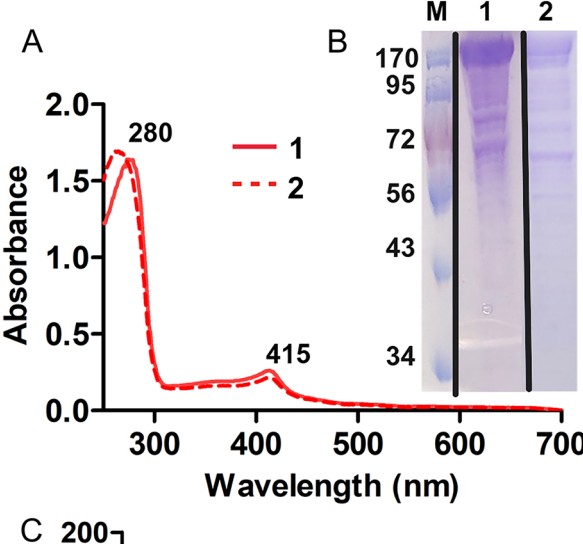

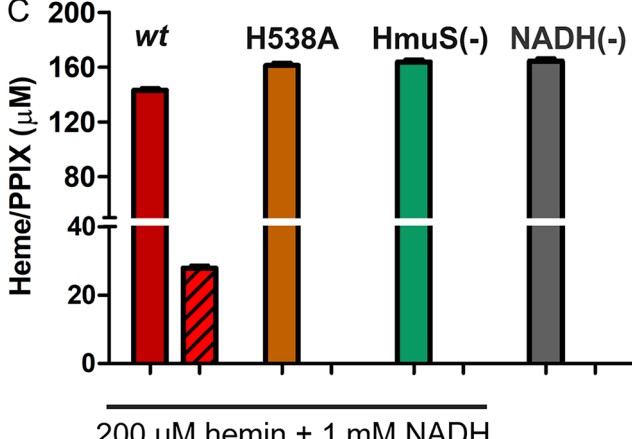

**Figure 6. H538A HmuS binds heme like wt HmuS but does not catalyze heme conversion to PPIX and Fe²⁺.**

(A) The UV/vis absorbance spectra for wt HmuS (solid line, 1) and the H538A mutant (dashed line, 2) indicate an oxidized, 6-coordinate heme protein following purification by anion exchange and size-exclusion chromatographies. The band centered at 280 nm is due to 20 μM protein. Pyridine hemochromagen analysis showed 2.3 μM heme content in wt HmuS and 1.8 μM heme in the H538A mutant. (B) SDS-PAGE of a molecular weight marker (M) and purified HmuS (lane 1) are reproduced from Fig. 3A for comparison with the mutant H538A HmuS (lane 2). Black lines indicate the interfaces of separate gel images aligned via the molecular weight markers. (C) Recombinantly expressed wt and H538A mutant HmuS proteins were examined for catalytic activity against heme. Reaction mixtures contained 200 μM hemin, 1 mM NADH and 20 μM wt HmuS (red bars) or H538A mutant HmuS (orange bars). No-protein (green bar) and HmuS/no-NADH (gray bar) controls were examined concurrently. Reaction mixtures were prepared in 20 mM Tris buffer, pH 7.1 (total volume 330 μL) and incubated anaerobically at 25 °C for 30–40 min prior to extraction and HPLC analysis for unreacted heme (solid bars) and PPIX product (striped bars). Triplicate results were averaged. Error bars represent standard deviation. Source data are available online for this figure.

specifically in a selection of 680 HmuS sequences, but not in the CobN/ChlH sequences. The same axial ligand set is common in members of the cytochrome c family of electron transporters (Hüttemann et al, 2011). We assign this as the non-substrate heme observed by UV/visible absorbance spectroscopy in the as-isolated HmuS ($Kd = 6.0 \pm 6\ \mu M$) (site 1).

**Figure 7. Minimal mechanism for heme de-chelation by HmuS.**

Possible domain locations for each step are proposed. Source data are available online for this figure.

In addition, 34 Å away and inside the body's cavernous interior, two cations (assigned as Na1 and Na2) were identified amid a central cleft lined by highly conserved residues. This region (site 2) features a strategically well-conserved mix of hydrophobic and hydrophilic surfaces that readily accommodate PPIX docked with Vina. Importantly, the docked PPIX is centered on Na1, suggesting Na1 serves as a proxy for Fe(II) and also marks a potential location for substrate conversion to products, which then travel outward from the protein (Fig. 5). The modeled site includes His-1209/Asp-1208 and His-538/Glu-577 histidine-carboxylate dyads which could each act as an axial ligand or acid-base residue, and Ser/Gln residues and main chain amines at the *N*-terminus of helix V-α3 predicted to interact with propionates (Fig. EV5). Substitution of the cis-non-prolyl His-538 with alanine preserved heme binding in the spectroscopically and structurally observed site, but completely eliminated heme conversion to PPIX, strongly implicating the Na1 location (site 2) as the site of metal de-chelation.

The enzyme's dependence on NADH, the lack of detectable porphyrin-derived products other than PPIX, and the spectroscopically observed reduction of the HmuS-bound heme (Figs. 2 and 3) together suggest that NADH reduces the site 1 heme iron, though we do not yet know how. Like many bacteria, including *E. coli*, *B. theta* encodes at least one flavodoxin that is highly upregulated under iron-deficient conditions and that could meter an NADH-derived electron to the heme iron (Lewis and Gui, 2023). (An *E. coli* reductase was identified in minuscule amounts by mass spectrometry in the ion-exchange-purified HmuS but not appreciably in the preparation following size-exclusion chromatography; see Dataset EV2.) The flexible hinge connecting the head domain to the neck and the proximity of the MRI suggest that ferric heme can access a binding site at the head/neck interface, possibly fed by the MRI. This could be the site of heme-iron reduction or receipt of a reduced heme. Though the site 1 heme must be in the reduced state for PPIX production to occur, the role of this heme is currently unclear and under investigation.

Evolutionarily and structurally unrelated ferrochelatases catalyze the same reaction as HmuS, though biased in the reverse direction ($Fe(II) + PPIX^{2-} \leftrightarrows [Fe(II)PPIX]^{2-} + 2H^+$) (Dailey et al, 2000; Obi et al, 2022). The well-studied ferrochelatase paradigm and the experimental observations presented here suggest a model for how HmuS might demetallate heme at site 2 (Fig. 7). Deformation of the planar PPIX ligand would expose the pyrrole nitrogens (Shi et al, 2005) to proton-donating side chains, either from alternate faces of the porphyrin plane or from the same face with both pyrrole nitrogens aspiring toward the same active site acid. Protonation of these nitrogens is expected to restore planarity to the deformed PPIX as $Fe^{2+}$ is liberated from the macrocycle. A conserved pathway for metal egress, leading to the inner membrane $Fe^{2+}(aq)$ transporter (FeoB), could originate at the Na2 site (Marlovits et al, 2002). A defined pathway for PPIX release is likely. The entry/exit of substrates and products could involve repositioning flexible, highly conserved Loop 1 (Fig. 5), which lacks secondary structure and assumes a different configuration in ChlH.

## Chelatase superfamily

Confirming the function of HmuS as a *de-chelatase* has also motivated redrawing the boundaries of the sequence superfamily and subfamilies. Alignment of HmuS sequences derived from complete *hmu* operons supplied a core set of HmuS-family-defining sequence features, including the MRI, Met-79, and His-254, and the series of motifs within the HmuS body described above. Aligning the much larger superfamily of sequences, including the HmuS set as well as probable CobN and ChlH sequences, identified a far shorter list of residues conserved superfamily-wide, placing HmuS distinctions in even sharper relief (Appendix Figs. S11–S13; Dataset EV3). To perform a deeper analysis of the chelatase superfamily, we combined search results for representative members of the three major families. After converting these ~10,000 sequences into complex numerical representations using protein language models (ProtT5), a 2D representation of the whole dataset was created by t-distributed stochastic neighbor embedding (t-SNE) (van der Maaten and Hinton, 2008; Elnaggar et al, 2022). Subdivision of the entire superfamily is not done just according to sequence similarity, because it also includes a complex encoding of the physical and chemical properties of individual sequences. In this analysis, HmuS sequences form an island amid an archipelago containing more than the minimally anticipated 3 familial clusters (Fig. 8; Dataset EV4). Biochemically, genetically, and/or structurally characterized representatives of these groups functionally anchor some of the clusters. The existence of further subdivisions beneath these known functional umbrellas, in some cases, appears to follow taxonomic lines; for example, green plants and phototrophic cyanobacteria possess $Mg^{2+}$ chelatases whose sequence divergence may

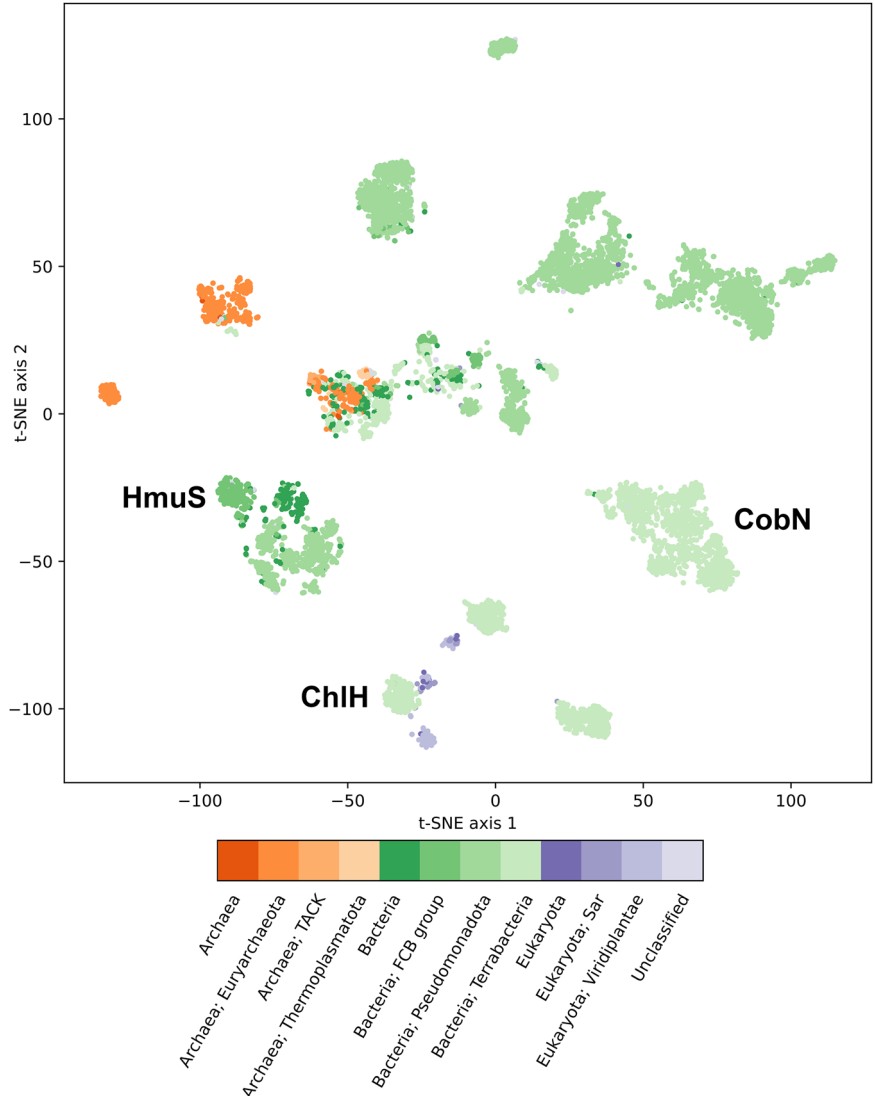

**Figure 8. 2D embedding of chelatase superfamily sequences, illustrating probable family membership of HmuS, CobN, ChlH, and other (>10,000) mutually homologous sequences.**

High-dimensional numerical representations of protein sequences were reduced to two dimensions by t-SNE (van der Maaten and Hinton, 2008). Each data point was colored by its taxonomic domain membership: Archaea (orange), Bacteria (green), and Eukarya (purple). Phyla with the largest memberships within these domains are colored separately in shades of a similar color. Representative clusters for each of the three major families are labeled. Family assignments (HmuS, ChlH, and CobN) are made on the basis of biochemically characterized anchoring sequences, of which there are very few. Only the clusters nearest to the labels on the diagram can be ascribed a known function. Most of the remaining bacterial and archaeal sequences cannot be reliably annotated without further investigation. See Dataset EV4 for an HTML-enabled representation of these data, in which the identity of individual points can be visualized by hovering the mouse over each. Source data are available online for this figure.

reflect the distance between the taxa. Other sequences, currently annotated as CobN subunits, come predominantly from Methanobacteriota and cluster according to taxonomic class into sibling groups from *Halobacteria* and thermophilic or methanogenic eubacteria, though they are not associated with other *cob* genes and their functions are unclear.

We were surprised to find so many members of the parent superfamily in taxonomically distant groups, sometimes with multiple paralogs in the same species. We wondered whether proteins like HmuS might have biological roles outside the *hmu* pathway. Salvage pathways using chelatase/de-chelatase processes might be important, for example, for repurposing the PPIX

generated by heme catabolism in the microbiome environment. Alternatively, metallation/demetallation could permit activation and deactivation of cofactors on an as-needed basis. Chlorophyll, for example, is known to be reversibly demetallated and converted to pheophytin by protonation, or remetallated with Zn(II) or Cu(II) to stabilize it against heat (Tanaka and Ito, 2025). These reactions, which are catalyzed by enzymes that are not homologous to HmuS, offer a mechanism for photosynthetic organisms to protect themselves or adjust their photosynthetic efficiency according to diurnal or seasonal cycles, without sacrificing a metabolically expensive cofactor. Metallation/demetallation could likewise be a means by which microbiome species adjust their cellular heme

concentrations in response to shifting conditions. Alternatively, demetallation/remetallation could be essential steps toward further functionalization of ingested heme b around its perimeter, funneling it toward more specialized roles. Finally, a large number of HmuS homologs were identified in diverse species of Archaea. Methanogenic Archaea are known for their use of F430, a Ni(II)-dependent cofactor constructed from uroporphyrinogen: a central metabolic precursor of siroheme, F430, and heme b in many of these species. It is unclear how Archaea might use HmuS/CobN/ChlH homologs, particularly since these species do not possess complete *hmu* operons for heme assimilation, nor do they have aerobic pathways for generating chlorophyll or cobalamin.

## Protoporphyrin IX

Interestingly, the immediate products of heme de-chelation do not include CO (or formaldehyde) and consequently are not detected by the CO-sensitive circuitry that regulates human and microbial metabolisms and their interplay (Hopper et al, 2020). Moreover, Fe(II) and PPIX are toxic. While the lengths to which cells go to protect themselves from Fe(II)-driven redox stress and associated DNA, lipid, and protein damage, especially near aerobic/anaerobic interfaces, are well-known, the fate of PPIX remains to be discovered. It is possible that some PPIX is further metabolized by unknown mechanisms or retrieved by other organisms that rebuild it into heme. PPIX itself has also been identified in many pathogenic microenvironments. When exposed to light, PPIX fluoresces, offering a well-used means for endoscopically identifying porphyrin-associated pathogens like *Bacteroides fragilis* inside surgical and other necrotic wounds (König et al, 2000; Jones et al, 2020). Photoexcited PPIX facilitates the conversion of triplet $O_2$ to the highly reactive singlet state, which is leveraged in photo-dynamic therapies to eliminate pathogen-infected, cancerous, or other diseased tissue, and which leads to unwanted cell death in patients with PPIX-accumulating porphyrias. Porphyrins likewise hyperaccumulate in tumor microenvironments and associated serum, where they π-stack, intercalate into membranes, or precipitate. Because heme, PPIX, and iron are not appreciably excreted by healthy individuals, we expect that HmuS and allied pathways manage these critical molecules in a healthy microbiome, reaping their nutritional and metabolic benefits while mitigating their potential for harm.

## Conclusions

This work defines a long-sought, GI-tract-associated, microbial mechanism for anaerobically metabolizing heme. This mechanism is ubiquitous in human gastrointestinal metagenomes and abundant in other anaerobic ecosystems. It could be a principal, microbiome-assisted mechanism by which humans assimilate dietary and intestinally resorbed heme iron. The fate of the toxic PPIX byproduct, and the influence of this pathway on microbiome composition, are still unknown. Understanding HmuS function is essential to engineered approaches to directing microbiome-host metabolism and offsetting heme-associated pathologies beyond infection including anemia, inflammation, and cancer. HmuS is evolutionarily related to chelatase systems that catalyze metal insertion into chlorophyll ($Mg^{2+}$) and vitamin B12 ($Co^{2+}$), though HmuS works in the opposite, de-chelatase direction. We propose

that flexible metal chelation/de-chelation, using the widespread HmuS homologs identified informatically in this work, may play unknown roles in the metallobiochemistry of a variety of species.

# Methods

### Reagents and tools table

| Reagent or resource | Source | Identifier |
|---|---|---|
| **Bacterial and viral strains** | | |
| *Bacteroides thetaiotaomicron* V-5482 (wild-type) | ATCC | ATCC 29148 |
| *hmuS* mutant (1) *Bacteroides thetaiotaomicron* BT0494 | Arjes et al, 2022 | BT0494 P289-C05 |
| *hmuS* mutant (2) *Bacteroides thetaiotaomicron* BT0495 | Arjes et al, 2022 | BT0495 P295-G04 |
| *E. coli* BL21(DE3)-Lemo cells | New England Biolabs | Cat#C2528J |
| **Chemicals, peptides, and recombinant proteins** | | |
| Hemin chloride | Calbiochem® | Cat#3741 |
| Bathophenanthroline disulfonic acid disodium salt hydrate | Thermo Scientific™ | Cat#B23244.03 |
| B-Nicotinamide adenine dinucleotide reduced disodium salt | Thermo Fisher Scientific | Cat#AAJ6163803 |
| **Critical commercial assays** | | |
| Arkansas IDeA facility gel band proteomics analyses | | https://chemistry.uark.edu/research/statewide-mass-spectrometry-facility/ |
| University of Notre Dame Mass Spectrometry Facility bottom-up proteomics analyses | | https://massspec.nd.edu/ |
| **Deposited data** | | |
| HmuS structure | This paper | PDB: 9D26 |
| HmuS cryo-EM single particle map | This paper | EMDB: 46483 |
| **Experimental models: organisms/strains** | | |
| *Bacteroides thetaiotaomicron* V-5482 (wild-type) | ATCC | ATCC 29148 |
| **Recombinant DNA** | | |
| pET28a(+) containing hmuS gene from *Bacteroides thetaiotaomicron* V-5482 | GenScript | *HmuS* accession AE015928.1 |
| pET28a(+) | Snapgene | https://www.snapgene.com/plasmids/pet_and_duet_vectors_(novagen)/pET-28a(%2B) |
| **Software and algorithms** | | |
| CryoSparc | Punjani et al, 2017, 2020 | https://cryosparc.com/ |
| Phenix | Terwilliger et al, 2018 Pavel et al, 2013 Afonine et al, 2018 Liebschner et al, 2019 | https://phenix-online.org/ |
| SerialEM | Mastronarde, 2005 | https://bio3d.colorado.edu/SerialEM/ |
| SmartScope | Bouvette et al, 2022 | https://docs.smartscope.org/ |

| Reagent or resource | Source | Identifier |
|---|---|---|
| Coot | Emsley et al, 2010 | https://www2.mrc-lmb.cam.ac.uk/personal/pemsley/coot/ |
| Pymol | Delano, 2002 | https://pymol.org/ |
| Chimera | Pettersen et al, 2021 | https://www.cgl.ucsf.edu/chimera/ |
| ChimeraX | Meng et al, 2023 | https://www.cgl.ucsf.edu/chimerax/ |
| Cblaster | | https://github.com/gamcil/cblaster) |
| Identical Protein Groups database | | (https://www.ncbi.nlm.nih.gov/ipg/) |
| CD-HIT | | http://cd-hit.org |
| HH-blits | | https://github.com/soedinglab/hh-suite |
| Biorender | | Biorender.com |
| ClustalΩ | | https://www.ebi.ac.uk/jdispatcher/msa/clustalo |
| ProtT5 protein language | Elnaggar et al, 2022 | https://github.com/mheinzinger/ProstT5 |
| Open t-SNE | Poličar et al, 2024 | https://github.com/pavlin-policar/openTSNE |
| **Other** | | |
| Hypersil GOLD™ column (4.6 mm x 250 mm, 5 μm particle size) | Thermo Scientific™ | 25005-254630 |
| DEAE Sepharose Fast Flow | GE Healthcare | 17-0709-01 |
| Sephacryl™ S-200 High Resolution | GE Healthcare | 17-0584-01 |

## Experimental model

### Strains

The sequenced genome for the strain of *B. theta* used in this work (*Bacteroides thetaiotaomicron* V-5482, ATCC 29148, GenBank accession: AE015928.1) contains one complete *hmuYRSTUV* operon (loci (BT0497-BT0491)) with the *hmuS* gene on the complement strand at nucleotide positions 608,809–613,201 (Appendix Fig. S4). The original genome annotation erroneously assigned this gene to two loci (BT0495, BT0494), though they were subsequently identified as a single gene in all Bacteroidetes strains (1469 aa, RefSeq: WP_008765019.1) (Meslé et al, 2023). A transposon insertion library for *B. theta* V-5482 was generated by Arjes et al, 2022. Gene-inactivating, single transposon insertions into the *hmuS* coding region were identified in two of the library strains, mapping to nucleotide positions 610,214 and 612,897, respectively (Appendix Fig. S4). These mutants were obtained from the Huang group at Stanford University. The *wt* strain was acquired from the ATCC.

### Monitoring B. theta growth in chemically defined media

*B. theta* was cultivated in a pH buffered, chemically defined minimal medium (MM) composed of: 6.6 mM potassium dihydrogen phosphate, 15.4 mM NaCl, 98 μM $MgCl_2 \cdot (H_2O)_6$, 176.5 μM $CaCl_2 \cdot (H_2O)_2$, 4.2 μM $CoCl_2 \cdot (H_2O)_6$, 50.5 μM $Mn(II)Cl_2 \cdot (H_2O)_4$, 9.3 mM $NH_4Cl$, 1.75 mM $Na_2(SO_4)$, 134 μM L-methionine, 23.8 mM sodium bicarbonate, 8.25 mM L-cysteine (free base), 28 mM D-glucose, and 15 μM hemin chloride. The pH of the medium was adjusted to 7.1 using 5 M HCl. The MM supplemented with 15 μM hemin was transferred into Balch-type, crimp-topped tubes (10 mL MM) and kept in the anaerobic glove box for 4 h before use (2.5% $H_2$/97.5% $N_2$ atmosphere) (Bacic and Smith, 2008).

*B. theta* glycerol stocks were used to prepare the pre-inoculum in liquid MM + 15 μM hemin. After 14 h growth, the liquid pre-inoculum was centrifuged at $3260 \times g$ for 10 min at 4 °C. The cell pellet was washed twice and resuspended with MM without hemin, and used as an inoculum for monitoring bacterial growth and/or metabolite production. For optical monitoring of growth, MM (10 mL) in Balch-type, crimp-topped tubes containing 15 μM hemin with or without added 0.3 mM BPS was inoculated with sufficient cell mass to produce an initial optical density ($OD_{600nm}$) of 0.03 in the Balch tube (2.5% $H_2$/97.5% $N_2$ atmosphere), incubated at 37 °C, 150 rpm, and monitored every 2 h for 26 h. For larger-scale *B. theta* fractionation experiments, 70 mL cultures were grown in crimp-top bottles containing anaerobic MM medium supplemented with 15 μM hemin, prepared in the anaerobic chamber as described above.

## Method details

### Biochemical stock solutions and equipment

Chemicals were purchased through Fisher unless otherwise stated. Hemin (ferric heme chloride, Calbiochem®) stocks were prepared in 1 M NaOH (pH 13) or DMSO and concentrations measured via UV/Vis absorbance ($\varepsilon_{385} = 58.44$ mM$^{-1}$ cm$^{-1}$). In total, 200 μM PPIX stocks were prepared in acidified acetonitrile (ACN): ACN:1.7 M HCl (82:18, v/v) (da Silva et al, 2024). Both stocks were prepared in air and handled in amber bottles before storing at −20 °C. Diluted working solutions were prepared and used immediately (within 4 h). Reductant stocks (NADH, dithionite) were prepared in an anaerobic chamber (Coy Labs, 2.5% $H_2$ and 97.5% $N_2$) at 100 or 50 μM concentrations, respectively. Hemin and reductant stocks were aliquoted into single-use tubes, frozen, and discarded at the end of a day's use. Working solutions were similarly discarded within a day. An extraction solvent consisting of acetonitrile, 12 M HCl, and DMSO in a 41:9:50 volume ratio (final [HCl] = 1.08 M) was used to equivalently recover heme and PPIX from cells and reactions. Standards and reductants were diluted immediately prior to use. 20 mM TrisHCl buffer, pH 7.1 was used to resuspend *B. theta* cell pellets prior to lysis and subsequent analyses. The same buffer was used as a storage and reaction medium for purified, heterologously expressed HmuS, though with added 250 mM NaCl.

UV/vis spectra were measured in quartz cuvettes using a Cary 50 or in test tubes using a ThermoScientific Genesis instrument. Cell pellets (>1 g) were lysed via Branson model 102C (CE) sonicator. High-pressure liquid chromatography (HPLC) was performed on a Shimadzu Prominence-i LC-2030C 3D Plus with UV/vis detection (190–800 nm) using a Hypersil GOLD™ column (Thermo Scientific™, 4.6 mm × 250 mm, 5 μm particle size). Protein chromatography was carried out with a Next Generation Chromatography System (BioRad).

Analytical experiments were carried out with at least 3 replicates and averaged. Error bars are ±1 standard deviation.

### Graphics

Figure 1 was created with BioRender.com. Growth curves and bar charts were plotted using GraphPad Prism. UV/visible absorbance

data were plotted and fit using Kaleidagraph 4.0. Structural data were plotted using PyMol. Plotted/fitted data were uploaded into BioRender.com and amended with labels and legends to generate high-resolution, publication-quality graphics (Figs. 2 and 8 and graphical abstract).

### Extracting heme and PPIX from B. theta at the end of growth experiments

10 mL Balch tube cultures of *B. theta* cells were pelleted by centrifugation, washed twice in ultrapure water, resuspended (0.12 g mL$^{-1}$) in extraction solvent, transferred into FastPrep Lysis B-matrix tubes, and lysed using a FastPrep 24 5 g instrument (2 cycles: 6.0 m/s for 40 s). Samples were centrifuged (9600 × *g*, 25 °C, 15 min), yielding a well-resolved upper layer that was removed for quantifying PPIX and heme by HPLC (da Silva et al, 2024).

### Generating B. theta cell-free extracts

In all, 1 g pellets (on the order of $2.6 \times 10^{11}$ *B. theta* cfu) (Meslé et al, 2023) from 200 mL *B. theta* cultures grown as described above were resuspended in 15 mL 20 mM Tris-HCl buffer (pH 7.1) and lysed by sonication on ice (Branson instrument: 7 min, with a pulse sequence of 10 s on and 25 s off, at 40% amplitude). Whole lysates were dialyzed in air against the same buffer at 4 °C using 12 kDa molecular weight cut-off (MWCO) tubing (3 cycles of >3 h each). Soluble proteins were separated from the lysate by centrifugation (9600 × *g*, 25 °C), and contained 1.65 mg mL$^{-1}$ protein (estimated by the Bradford assay, below).

### Heme conversion to PPIX

All assays were carried out under $N_2$. 300 μL samples of the dialyzed lysate and soluble fractions were placed in Eppendorf tubes inside the glove box and amended to contain 1 mM NADH, 1 mM ATP, both, or neither. Reactions were initiated by adding 100 μM hemin (330 μL total reaction volume) and incubated in an anaerobic chamber at room temperature for 40 min prior to adding 300 μL extraction solvent, which denatured the proteins and solubilized both PPIX and unreacted hemin (da Silva et al, 2024). Proteins were precipitated by centrifugation (9600 × *g*, 25 °C, 20 min) and the organic layer removed for analysis by HPLC. To verify that hemin was not contaminated with PPIX or modified by incubation with the extraction solvent, hemin solutions were routinely analyzed by HPLC. Control reactions contained the same reactants and buffers but were stopped by the addition of the extraction solvent at time = 0. Hemin conversion to PPIX was not observed above the limit of detection in any case. The concentrations of heme and PPIX were expressed in micromolar (μM) based on the standard curve described below. All experiments were carried out in triplicate and averaged.

### High-performance liquid chromatographic (HPLC) analysis of heme and protoporphyrin IX

Chromatography was carried out using a linear gradient with solution A (ultrapure water + 0.1% trifluoroacetic acid, TFA) and solution B (ACN + 0.1% TFA), flow rate of 1 mL min$^{-1}$, oven temperature at 25 °C (Appendix Fig. S1). In all, 30 μL samples were injected onto the column. Standard curves of hemin and PPIX (0.1, 0.5, 1, 5, 10, 50 μM) were prepared to quantify the extracted heme and PPIX via peak integration.

### Heterologous expression of HmuS

The soluble HmuS protein from *B. theta* (accession WP_022471467.1) was heterologously expressed in *E. coli* BL21(DE3)-Lemo cells (NEB) without the predicted single transmembrane spanning helices at its N- and C-termini (Appendix Figs. S6 and S7). The gene was codon-optimized for *E. coli* and synthetically introduced between the NdeI/XhoI sites of pET28a(+) by GenScript (Appendix Fig. S6). The same plasmid was used to generate the H538A mutant of HmuS, also by GenScript.

For protein expression, a single colony from a freshly streaked Lysogeny Broth (LB)-agar plate was used to inoculate an overnight liquid LB culture on a rotary shaker (37 °C, 200 rpm). This subsequently seeded $6 \times 1$ L flasks containing terrific broth. All media contained selective antibiotics (50 μg mL$^{-1}$ kanamycin, 34 μg mL$^{-1}$ chloramphenicol). Flasks were incubated on a rotary shaker to mid-logarithmic growth (OD$_{600}$ = 0.6-0.8). Isopropyl β-D-1-thiogalactopyranoside (IPTG, 1 mM) was added to induce protein expression at 16 °C, 200 rpm, for 16 h. The cultures were pelleted by centrifugation and stored at –80 °C. Cell pellets were resuspended in 100 mL of 20 mM Tris pH 7.1 and lysed by ultrasonication on ice with added protease inhibitors (0.5 mM phenylmethylsulfonate, 1.0 mM EDTA, and a protease inhibitor tablet (Protease Inhibitor Tablets, EDTA-free, Thermo Scientific). The lysate was ultracentrifuged (39,000 × *g*, 30 min, 4 °C) to obtain a clear pink supernatant (clarified lysate).

### HmuS purification

Working in ambient air at 4 °C, the clarified lysate (100 mL) was loaded onto 80 mL HiPrep DEAE FF 16/10 DEAE sepharose fast flow anion exchange resin (Cytiva) equilibrated with 20 mM Tris pH 7.1. The column was washed with 800 mL 20 mM Tris-HCl and eluted using a linear gradient of the same buffer containing 0–500 mM NaCl (3 mL min$^{-1}$, 70 min). Fractions were analyzed by SDS-PAGE (12% acrylamide) and UV/visible absorbance spectroscopy. Fractions enriched ≥50% in a protein of the expected molecular weight (158 kDa) were pooled. This fraction was centrifuge concentrated (100 kDa MWCO) to ≤1 mL and loaded onto a 140 mL size-exclusion column (HiPrep 16/60 Sephacryl S-300 HR, Cytiva) pre-equilibrated with 20 mM Tris pH 7.1, 250 mM NaCl. Proteins were eluted isocratically in the same buffer at 0.1 mL min$^{-1}$. HmuS-enriched fractions were identified by SDS-PAGE, concentrated, flash frozen in liquid $N_2$, and stored at –20 °C, ≥10 mg mL$^{-1}$ until further use.

### Protein concentrations

Protein concentrations were determined by Bradford assay. 0.1 mg mL$^{-1}$ bovine serum albumin (BSA from BioRad, stock concentration 2 mg mL$^{-1}$) was used to make 2.5, 5, and 10 μg mL$^{-1}$ standards. Absorbance at 595 nm was recorded (corresponding to the protein-dye complex). Standards and unknowns were recorded in triplicate and average concentrations reported.

### Heme concentrations by pyridine hemochromagen assay

The heme content of purified HmuS was measured by mixing 400 mL HmuS-heme complex (10 mg mL$^{-1}$) with 400 mL 40% pyridine in 0.2 M NaOH(aq) along with 2 mL 0.1 M potassium ferricyanide and measuring the absorption spectrum of the oxidized *bis*-pyridine-bound heme. The oxidized heme-pyridine complex was then reduced by 20 μL 0.5 M sodium dithionite in

0.5 M NaOH (excess), and the characteristic absorption spectrum of the reduced heme-pyridine complex was obtained with a Soret band at 420 nm and β/α-bands at 525 and 557 nm. The concentration of the bis-pyridyl ferrous heme was determined using the extinction coefficient at 557 nm ($\varepsilon_{557} = 34.7$ mM$^{-1}$ cm$^{-1}$) corrected for dilution. The [heme] in HmuS-heme samples determined by this method were used in assigning an approximate $\varepsilon$-value for the complex at its Soret peak maximum (Barr and Guo, 2015).

### Heme binding to HmuS

We attempted to remove heme from the as-isolated HmuS-heme complex by dialysis against an imidazole-containing buffer. Before dialysis, the absorption spectrum of the heme-bound purified HmuS protein (4 mg mL$^{-1}$) was recorded to compare with the spectrum of the dialyzed protein. The protein (2.5 mL) was dialyzed in 12–14 kDa MWCO dialysis tubing against 20 mM Tris-HCl pH 7, containing 50–300 mM NaCl and 50 mM imidazole for 6 h (two cycles each of 3 h), followed by two more cycles of dialysis against buffer without imidazole. The dialyzed protein was then concentrated to the starting concentration of 4 mg mL$^{-1}$ using a 100 kDa MWCO centrifuge concentrator. The absorption spectra (Soret position and intensity) pre- and post-dialysis were unchanged, indicating that the heme remained attached to HmuS.

To monitor ferric heme binding to purified, partially heme-bound HmuS, a freshly prepared aqueous solution of 144 μM hemin (pH 8) was added in small increments to 600 μL purified HmuS (0.95 mg mL$^{-1}$; 6 μM HmuS containing 0.6 μM HmuS-heme complex unless otherwise stated in the text) and monitored by UV/Vis absorbance spectroscopy. Samples were allowed to come to equilibrium after each addition. The final spectra in the experiments were unchanged after 1 h. A cuvette containing 600 μL buffer was titrated in parallel, and the difference spectra are reported. Ligand binding data were fit to a quadratic model since heme could not be used in large excess over the protein:

$$\Delta A = \Delta A_{sat} \left( \frac{[E] + [L]_T + K_d - \sqrt{([E] + [L]_T + K_d)^2 - 4[E][L]_T}}{2[E]} \right).$$

To saturate one or more binding sites on the protein, HmuS (500 μL, 10 mg mL$^{-1}$) was incubated with excess hemin, aerobically and in the dark. Unbound or loosely bound heme was removed by three cycles of centrifuge concentration (50 kDa MWCO) and resuspension in fresh buffer (20 mM tris-HCl, 250 mM NaCl, pH 7.1), and the UV/Vis absorbance spectrum measured. Concentrations of protein and heme in the sample following buffer exchange were measured by Bradford assay and pyridine hemochromagen assay, respectively.

### Monitoring heme turnover to PPIX by purified HmuS

Experiments paralleled those carried out with *B. theta* cell fractions. Reaction mixtures (total volume 330 μL) were prepared in an anaerobic glove box in 20 mM Tris-HCl buffer, pH 7.1 with 1 mM NADH and HmuS (wt or mutant; see text for concentrations/fractions assayed). In all, 100 μM hemin was added to initiate the reactions, which were incubated anaerobically (2.5% H$_2$/97.5% N$_2$ atmosphere) at room temperature for 40 min. Reactions were stopped and extracted by adding two volumes of extraction solvent. The proteins were pelleted by centrifugation, and the extracts

analyzed for heme and PPIX concentrations by HPLC (above). No-protein negative controls consisting of NADH and hemin were measured.

### Determining conservation of primary sequences

Using cblaster (https://github.com/gamcil/cblaster) against the Identical Protein Groups database (https://www.ncbi.nlm.nih.gov/ipg/), all HmuS proteins found in the same operon with the other 5 Hmu proteins were considered to fulfill the operonic criterion (Meslé et al, 2023). We selected only complete HmuS sequences with at least 95% coverage and at least 50% identity relative to *B. theta* HmuS. This approach yielded 1605 sequences. Using CD-HIT (http://cd-hit.org), this group of proteins was further cleaned to remove redundancies. The resulting 680 non-redundant HmuS operonic fasta sequences were aligned using ClustalΩ. Sequence logo representation was created at https://weblogo.threeplusone.com/create.cgi. To determine sequence conservation within the larger superfamily of chelatase proteins, we started with HmuS as query and performed 5 search iterations with HHblits (Remmert et al, 2012). Some of the identified ~1800 sequences were short, and we removed all sequences <1000 residues to generate a final alignment of 1513 chelatase sequences. See Dataset EV3 for spreadsheets containing alignment scores and structural representations of conservation.

### Clustering analysis

Sequences of *B. theta* HmuS, *M. tuberculosis* CobN (PDB code 7C6O) and *Synechocystis spp* magnesium chelatase ChlH (PDB code 6YT0) were used as queries for three iterations of PSI-BLAST searching. All hits were pooled together, and all redundant sequences were removed. After excluding sequences shorter than 1000 residues, 10,201 chelatase superfamily members were used for subsequent analyses. The ProtT5 protein language model was used to create an average 1024-value vector for each protein sequence (Elnaggar et al, 2022). All numerical vectors were subjected to dimensionality reduction by t-SNE to create a 2D grouping of sequences (Poličar et al, 2024). This type of non-linear embedding preserves the local relationships between vectors, but not necessarily their global distances. An interactive version of this embedding can be found as HTML in Dataset EV4. Hovering a mouse pointer over each dot will show its BLAST-determined sequence identity to *B. theta* HmuS, followed by UniRef annotation of that sequence.

### Cryo-EM single particle analysis

For cryo-TEM sample preparation, Quantifoil 300 mesh copper R 1.2/1.3 holey carbon grids were plasma cleaned for 45 s at 15 mA with a PELCO easiGlow discharge cleaning system (Ted Pella) and placed into a Vitrobot Mark IV (Thermofisher) blotting apparatus at 4 °C and 95% humidity with Whatman grade 1 blotting paper. 4 μL of HmuS at 1.5 mg mL$^{-1}$ was applied to each grid, blotted for 4 s at blot force 4, and immediately vitrified by plunge-freezing into liquid ethane. Upon freezing, the samples were clipped into autogrids and stored in liquid nitrogen until analyzed.

For cryo-TEM data collection, prepared grids were loaded into a Talos Arctica G2 transmission electron microscope (Thermofisher Scientific) operating at 200 kV and tuned for parallel illumination. Micrographs were collected using a Gatan K3 camera and a total electron exposure of 56e$^-$ Å$^{-2}$ distributed over 51-frame

dose-fractionated movies with a 3.06 s exposure time. SmartScope-controlled SerialEM was used to collect 15,994 exposures at a nominal magnification of 45,000 (0.9061 Å pixel$^{-1}$) with target defocus values from −0.6 to −1.5 μm using a $5 \times 5$ multi-shot scheme, or beam-image-shift (BIS) distance of 7.5 μm, where appropriate (Mastronarde, 2005; Bouvette et al, 2022). Data collection and initial processing were done with CryoSPARC Live. The initial viewing direction distribution showed a slight preferred orientation problem. To overcome this, we increased the number of 0° tilt angle exposures to 7800 and then collected an additional 8194 exposures with a 15° tilt angle.

Single particle reconstruction was performed with cryoSPARC (Punjani et al, 2017; Tan et al, 2017); the workflow is presented schematically in Appendix Fig. S9. Each recorded movie underwent patch motion correction and patch CTF estimation in cryoSPARC Live using default parameters. In all, 3392 movies were discarded based on quality and resolution of the CTF fit, calculated defocus, total motion, and relative ice thickness, leaving 12,602 for subsequent work. The cryoSPARC blob picker was used to pick 18,955,798 particles using a circular 60–120 Å blob size, an NCC score above 0.3 and local power between −76 and 1301. Particles were extracted with a box size of 416 pixels and binned by 2 (final box size = 208 pixels). The first 100,000 particles were used for an initial 3-class ab initio reconstruction. The full particle set was then washed twice using heterogeneous refinement seeded with the three volumes from the multiclass ab initio. The remaining 5,929,897 particles were re-extracted with an unbinned box size of 320 pixels and used for homogeneous refinement with default parameters. This was followed by (i) global CTF refinement while fitting tilt, trefoil, tetrafoil and anisotropic magnification, (ii) homogenous refinement, (iii) local CTF refinement, and (iv) non-uniform refinement (4), ultimately giving a reconstruction at 2.58 Å, as judged by gold standard Fourier shell correlation (FSC) with a 0.143 cut-off.

Subsequent efforts to resolve the MRI with focused 3D classification/refinement were unsuccessful. In contrast, a 3D classification job focused on domain 1 with the number of classes set to 4, gave 3 classes with no or extremely poor density for domain 1, but a 4th class (2,133,842 particles) with strong, well-defined domain 1 density. The three classes lacking domain 1 density were then combined (3,795,134 particles), and each of the particle subsets (ordered domain 1, disordered domain 1) were subjected to an additional round of homogenous refinement. Both maps showed strong density for domains 2–6. The 2,133,842 particle set refined to a global resolution of 2.60 Å by GSFSC and again showed strong domain 1 density, while the 3,795,134 particle set refined to 2.55 Å and lacked domain 1 density.

CryoSPARC orientation diagnostics (Appendix Fig. S10) reported a broad distribution of azimuthal tilt angles for each dataset. Thus, the minor preferred orientation problem noted during initial data collection was successfully overcome with the increased number of micrographs collected at two different tilt angles (0°, 15°). Consistent with this, the 3D FSC analysis (Tan et al, 2017) gave a conical FSC Area Ratio (cFAR) of 0.68 and minimum and maximum directional resolutions of 2.84 and 2.53 Å for the ordered domain 1 structure. Similarly, the disordered domain 1 structure had a cFAR of 0.72 with minimum and maximum resolutions of 2.45 and 2.76 Å. CryoSPARC documentation suggests a cFAR value above 0.5 serves as a reasonable

threshold for the lack of preferred orientation. In addition, the Fourier sampling analysis (Baldwin and Lyumkis, 2020; Baldwin and Lyumkis, 2021) implemented in cryoSPARC yielded Sampling Compensation Factors (SCF) of 0.891 and 0.864 for the respective ordered and disordered structures, where values above 0.81 generally indicate good, though not necessarily isotropic signal content.

### Structural model building and refinement

A homology model was generated (AlphaFold3) (Abramson, 2024) and docked into a sharpened ordered domain 1 map with Phenix (phenix.autosharpen, phenix.dock_in_map) (Terwilliger et al, 2018). The docked model was then rigid body fit by domain (phenix.real_space_refine) (Pavel et al, 2013; Afonine et al, 2018), and again using 20 smaller segments identified by the TLS Motion server. The model was then completed using iterative building in Coot (Emsley et al, 2010) and real-space refinement (phenix.real_space_refine) in Phenix (Afonine et al, 2018; Liebschner et al, 2019). Phenix real-space refinement included global minimization, local grid search, atomic displacement parameters, along with secondary structure, Ramachandran and rotamer outlier restraints. Occupancy refinement for the heme group and sodium atoms suggests these ligands are present at full occupancy (1.0). The model was deposited in the Protein Data Bank with accession code PDB ID 9D26 and the map was deposited in the Electron Microscopy Data Bank with accession code EMD-46483. Metrics for the ordered domain 1 model and map validation are presented in Appendix Table S3.

Domain 1 and heme were then deleted from the above model, and rigid body fit into the disordered domain 1 map. Further refinement of the disordered, heme-free structure then proceeded as described for the ordered domain 1 structure. The model was deposited in the Protein Data Bank with accession code PDB ID 9P4S and the map was deposited in the Electron Microscopy Data Bank with accession code EMD-71280. Metrics for the disordered domain 1 model and map validation are also presented in Appendix Table S3. Figures were prepared with Pymol (Delano, 2002), Chimera and ChimeraX (Pettersen et al, 2021; Meng et al, 2023). Protoporphyrin IX was docked to the HmuS structure using Autodock Vina as previously described (Trott and Olson, 2010).

### Proteomics analysis of excised SDS-PAGE gel bands

The identities of the expressed protein (158 kDa), its major contaminant band (60 kDa), and a band at 45 kDa which was part of the IEC fraction but later removed by SEC purification were verified by mass spectrometry (Arkansas IDeA facility). SDS-PAGE gel bands were excised and subjected to in-gel trypsin digestion. Gel segments were destained in 50% methanol (Fisher), 50 mM ammonium bicarbonate (Sigma-Aldrich), followed by reduction in 10 mM Tris[2-carboxyethyl]phosphine (Pierce) and alkylation in 50 mM iodoacetamide (Sigma-Aldrich). Gel slices were then dehydrated in acetonitrile (Fisher), followed by the addition of 100 ng porcine sequencing grade modified trypsin (Promega) in 50 mM ammonium bicarbonate (Sigma-Aldrich) and incubation at 37 °C for 12–16 h. Peptide products were then acidified in 0.1% formic acid (Pierce).

Tryptic peptides were separated by reverse phase XSelect CSH C18 2.5 mm resin (Waters) on an in-line 150 × 0.075 mm column using a nanoAcquity ultrahigh-pressure liquid chromatography

(UPLC) system (Waters). Peptides were eluted using a 60 min gradient from 98:2 to 65:35 buffer A:B ratio. [Buffer A = 0.1% formic acid, 0.5% acetonitrile; buffer B = 0.1% formic acid, 99.9% acetonitrile.] Eluted peptides were ionized by electrospray (2.4 kV) followed by MS/MS analysis using higher-energy collisional dissociation (HCD) on an Orbitrap Fusion Tribrid mass spectrometer (Thermo) in top-speed data-dependent mode. MS data were acquired using the FTMS analyzer in profile mode at a resolution of 240,000 over a range of 375 to 1500 *m/z*. Following HCD activation, MS/MS data were acquired using the ion trap analyzer in centroid mode and normal mass range with precursor mass-dependent normalized collision energy between 28.0 and 31.0. Proteins were identified by database search using Mascot (Matrix Science) with a parent ion tolerance of 3 ppm and a fragment ion tolerance of 0.5 Da. Scaffold (Proteome Software) was used to verify MS/MS based peptide and protein identifications. Peptide identifications were accepted if they could be established with less than 1.0% false discovery by the Scaffold Local FDR algorithm. Protein identifications were accepted if they could be established with less than 1.0% false discovery and contained at least 2 identified peptides. Protein probabilities were assigned by the Protein Prophet algorithm (Nesvizhskii et al, 2003). Label-free quantitation (LFQ) using signal intensities for average-of-3-most-abundant-peptides per protein, normalized to the number of identifiable peptides for a given protein, was performed using the MassQuant software. The resulting iBAQ metric was computed for the most abundant proteins (Appendix Fig. S7) (see Dataset EV1).

### Bottom-up proteomics data measured from IEC and IEC-SEC purified HmuS

To identify all contaminants, including a possible co-purifying reductase or other significant *E. coli* proteins, protein identities and relative abundances were assessed for IEC and IEC-SEC purified HmuS (University of Notre Dame Mass Spectrometry Facility). The yellow 20–30 kDa contaminant fractions isolated by SEC were also pooled, centrifuged, concentrated (10 kDa MWCO), and analyzed. Samples were dissolved in water at a total protein concentration of 1 µg/µL. Proteins were reduced (100 mM dithiothreitol, MP Biomedicals, LLC) in 25 mM ammonium bicarbonate then heated for 30 min, 65 °C. Solutions were cooled to room temperature then alkylated in the dark for 30 min following the addition of iodoacetamide (Sigma) (40 mM). Trypsin digestion proceeded overnight at 37 °C following the addition of 25 µL of 0.1 ng µL⁻¹ Trypsin Gold (Promega) in 25 mM ammonium bicarbonate. In all, 50 µL of digested solution was desalted via 5 passes of 10 µL using C18 Tips (Pierce). Fractions eluted in 50:50 water:acetonitrile with 0.1% trifluoroacetic acid were evaporated to dryness on a speedvac. Dried samples were reconstituted with 20 µL of 96:4 water:-acetonitrile with 0.1% formic acid for subsequent injection on the nanocolumn.

Overall, 1 µL of tryptic peptides was resolved with nano-scale ultrahigh-pressure liquid chromatography (nanoUPLC) coupled with tandem mass spectrometry using a system composed of a Waters M-Class in-line with a Q-Exactive HF mass spectrometer (Thermo Scientific). Solvent A (0.1% formic acid in water) and solvent B (0.1% formic acid in acetonitrile) were used as the mobile phase. Peptides were eluted from an Acquity BEH C18 Column, 1.7-µm particle size, 300 Å (Waters) column (100 µm inner diameter × 100 mm long) using a 48-min gradient at a flow rate

of 0.9 µL/min (4% B for 8.1 min, 4–7% B 10.0 min, 7-33% B 10–30 min, 33–90% B 30–33 min, 90% B for 3 min, 90–4% B 36–37 min, 4% B 37.1-48 min to equilibrate the column). Data were collected in positive ionization mode. Peaks Online (Bioinformatics Solutions) software was used to identify proteins from tandem mass spectra of peptide ions. The *E. coli* K12 protein database was modified to include the sequence of the expressed HmuS protein. Two missed cleavages by trypsin were permitted in the database search which included carbamidomethylation (C) as a fixed modification along with oxidation (M) and deamidation (N,Q) as variable modifications. The mass tolerance for precursor ions was set to 0.010 Daltons, and mass tolerance for fragment ions set to 0.8 Da (see Dataset EV2).

## Data availability

The cryo-EM structural model for HmuS was deposited in the Protein Data Bank with accession code PDB ID 9D26 and the map was deposited in the Electron Microscopy Data Bank with accession code EMD-46483. The model for the "headless" structure was deposited with accession code PDB ID 9P4S.

The source data of this paper are collected in the following database record: biostudies:S-SCDT-10_1038-S44318-025-00563-5.

## Peer review information

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

## Acknowledgements

AKN, RRS, EA, and JLD thank the National Institutes of Hea76lth grant R35 GM136390 (US Health and Human Services Department) for funding. Funding for the Montana State University Cryo-EM Core Facility (*RRID*:SCR_026324) was contributed by the National Science Foundation (DBI-1828765), the MJ Murdock Charitable Trust, the National Institute of General Medical Sciences (P30GM140963), and the MSU Office of Research, Economic Development and Graduate Education. Structure calculations and other analyses were performed on the Tempest High Performance Computing System, operated and supported by University Information Technology Research Cyberinfrastructure at Montana State University. We thank Dr. Jessica Lusty Beech and Victoria Adedoyin for technical support and Garrett Moraski for helpful discussions. We regret not being able to cite all of the relevant work in this field due to space limitations.

## Author contributions

**Arnab Kumar Nath**: Conceptualization; Resources; Data curation; Formal analysis; Supervision; Funding acquisition; Validation; Investigation; Visualization; Methodology; Writing—original draft; Project administration; Writing—review and editing. **Ronivaldo Rodrigues da Silva**: Conceptualization; Formal analysis; Validation; Investigation; Visualization; Methodology; Writing—original draft. **Colin C Gauvin**: Conceptualization; Data curation; Formal analysis; Validation; Investigation; Visualization; Methodology; Writing—original draft. **Emmanuel Akpoto**: Data curation; Formal analysis; Investigation; Methodology; Writing—original draft. **Mensur Dlakić**: Data curation; Software; Formal analysis; Validation; Investigation; Visualization; Methodology; Writing—original draft; Writing—review and editing. **C Martin Lawrence**: Conceptualization; Data curation; Software; Formal analysis; Supervision; Funding acquisition; Validation; Investigation; Visualization; Methodology; Writing—original draft; Project administration; Writing—review and editing. **Jennifer L DuBois**: Conceptualization; Resources; Data curation; Software; Formal analysis; Supervision; Funding acquisition; Validation; Investigation; Visualization; Methodology; Writing—original draft; Project administration; Writing—review and editing.

Source data underlying figure panels in this paper may have individual authorship assigned. Where available, figure panel/source data authorship is

listed in the following database record: biostudies:S-SCDT-10_1038-S44318-025-00563-5.

## Disclosure and competing interests statement

The authors declare no competing interests.

# Expanded View Figures

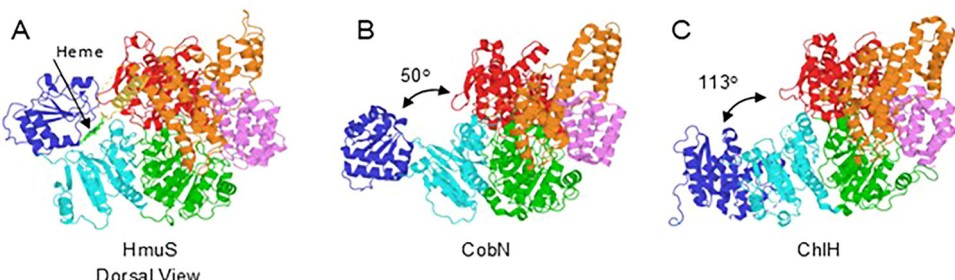

**Figure EV1. Head domain conformations.**

(**A**) HmuS is colored as in Fig. 4, with the head domain in blue, the neck in cyan, and domains III–VI in green, orange, red and purple. The orientation is a "dorsal" view, i.e., looking down on the domain IV "backbone", with the membrane roughly in the plane of the paper underneath HmuS, and heme bound at the head/neck interface. The first helix of the methionine-rich insertion (MRI) is shown in yellow, with the dashed line indicating a rough position for the 120 disordered residues of the MRI, potentially in position to interact with the heme-binding site at the neck/head interface. (**B**) CobN shown in an equivalent orientation. Domains II-VI superpose well on HmuS, but the head domain is rotated away from the neck domain by 50°. (**C**) ChlH in the same relative orientation, showing even greater movement of the head domain relative to the neck and body domains. Superposition of the HmuS head domain on the ChlH head domain requires a 113° rotation. The differing orientations of the head domain in CobN and ChlH suggest the HmuS head domain may also sample open conformations in the absence of heme.

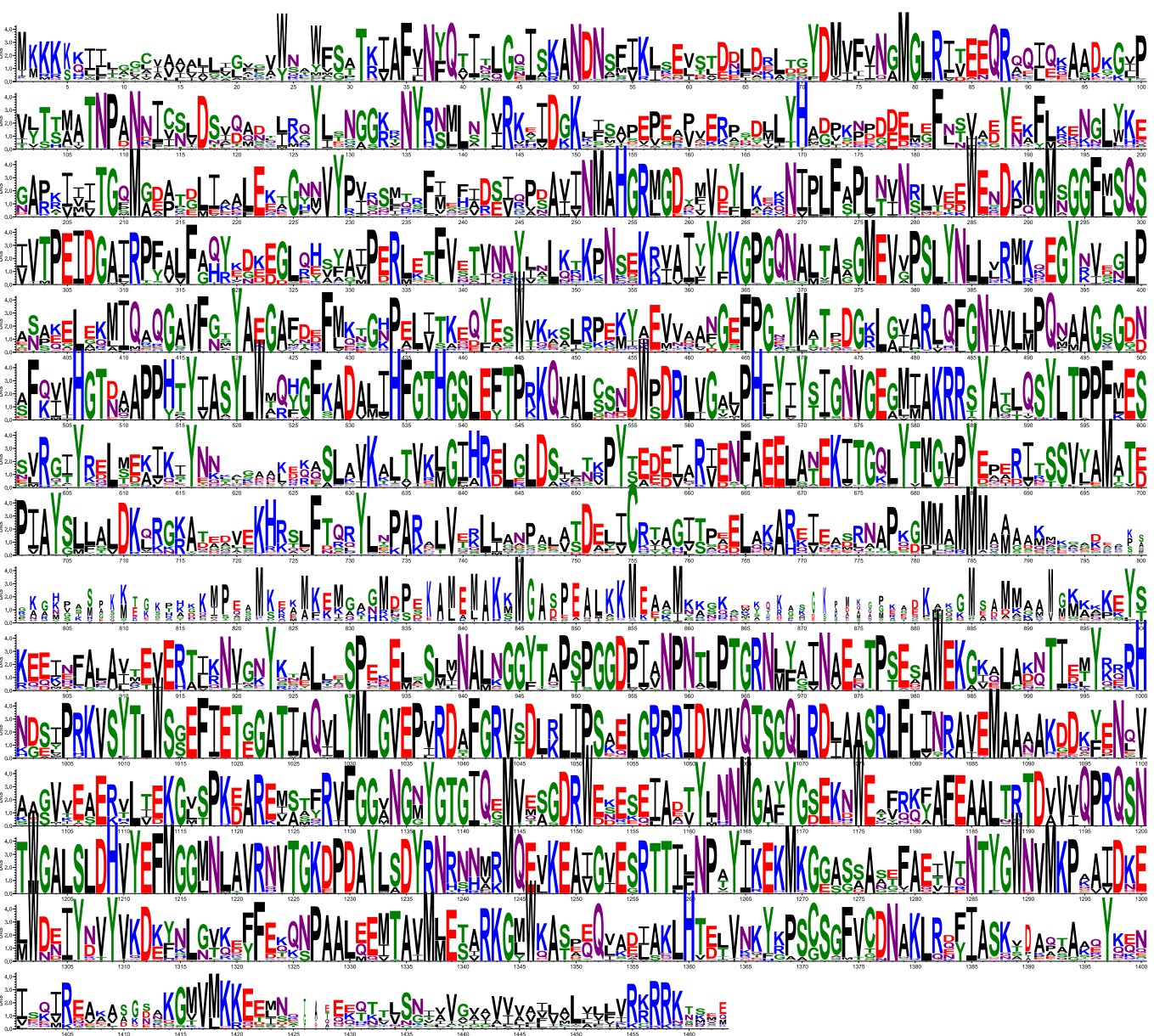

**Figure EV2. Sequence logo representation illustrating conservation among operonic HmuS sequences.**

As we included only operonic HmuS sequences with at least 50% identity and 95% coverage to *B theta* HmuS, this logo is most representative of organisms related to *B. theta*. See also Dataset EV4.

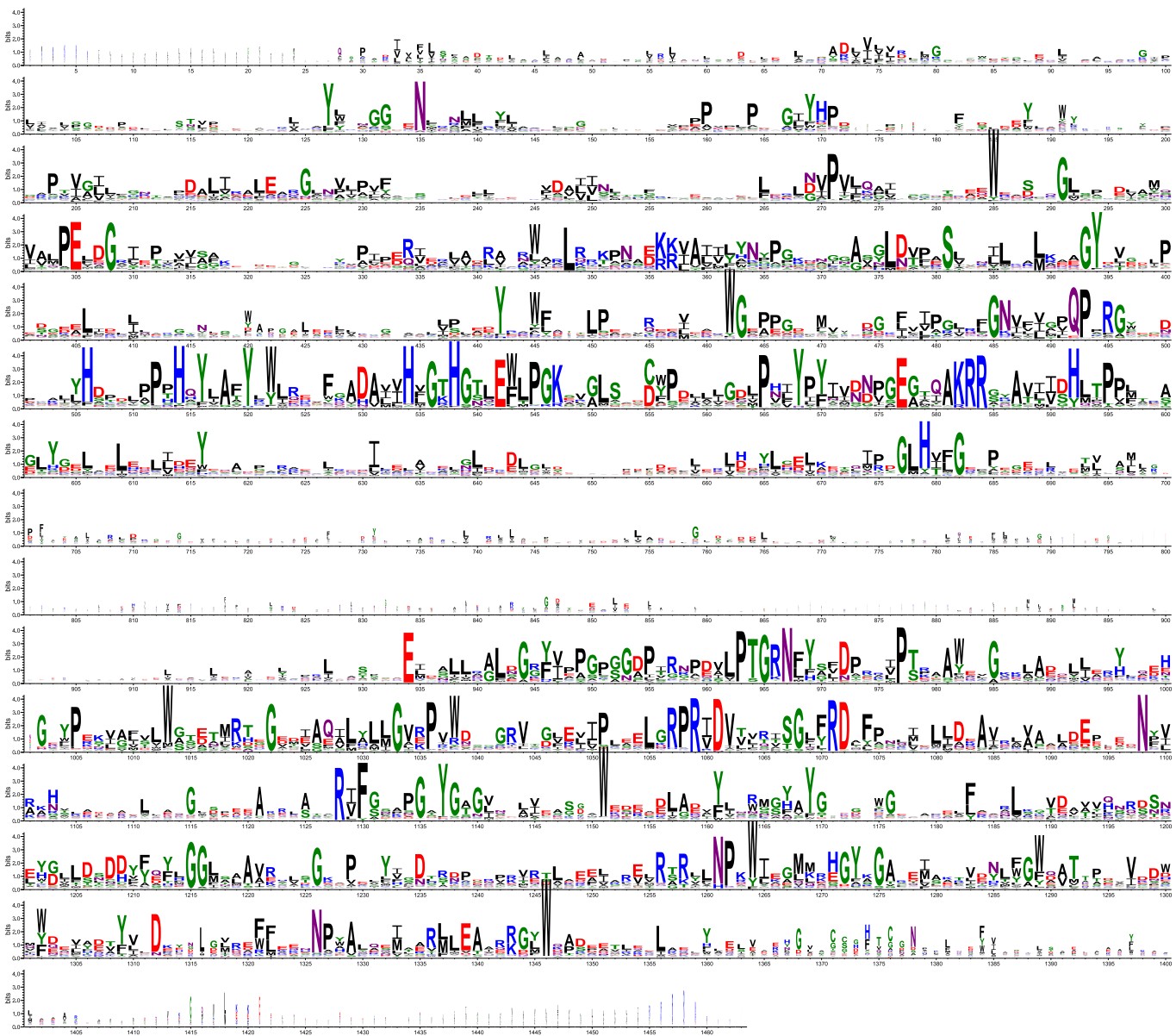

**Figure EV3. Sequence logo representation illustrating conservation among type 1 chelatase sequences.**

As the MRI is not found in chelatase sequences other than HmuS, the middle part of the sequence logo has almost no conservation.

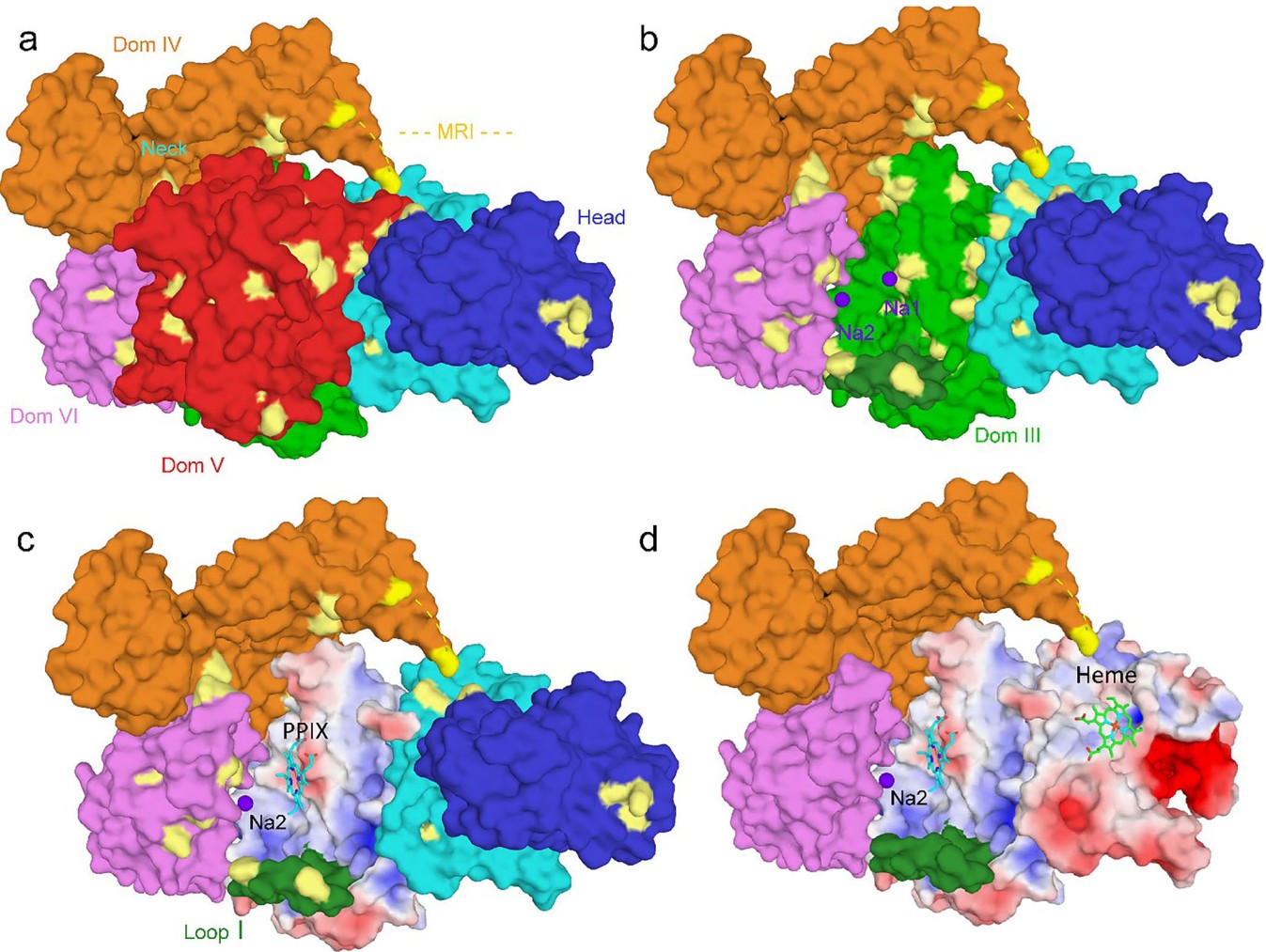

**Figure EV4.  Ligand binding sites, backside view.**

(A) HmuS is shown from the "backside", a 180° rotation about the vertical axis relative to Fig. 6. Domains are colored as in Figs. 5 and 6, with strictly conserved residues in pale yellow. The absence of the MRI is indicated by the yellow dashed line and yellow anchor points on domain IV (orange). (B) Domain V has been removed, revealing the central cleft with 2 bound Na ions (purple), Na1 and Na2, and Loop I at the bottom of the structure. Na1 is coordinated by strictly conserved His-1209, which lies in the center of the exposed domain III face. The domain III face of the central cleft is less conserved than the domain V face. (C) The docked protoporphyrin IX (PPIX) and Na2 are shown, along with the electrostatic surface domain III ($+/-$ 5kT/e). (D) Domain I (blue) is also removed, revealing the heme-binding site at the Head/Neck interface. The heme and PPIX molecules are separated by nearly 30 Å. Electrostatic surfaces are also shown for domain II.

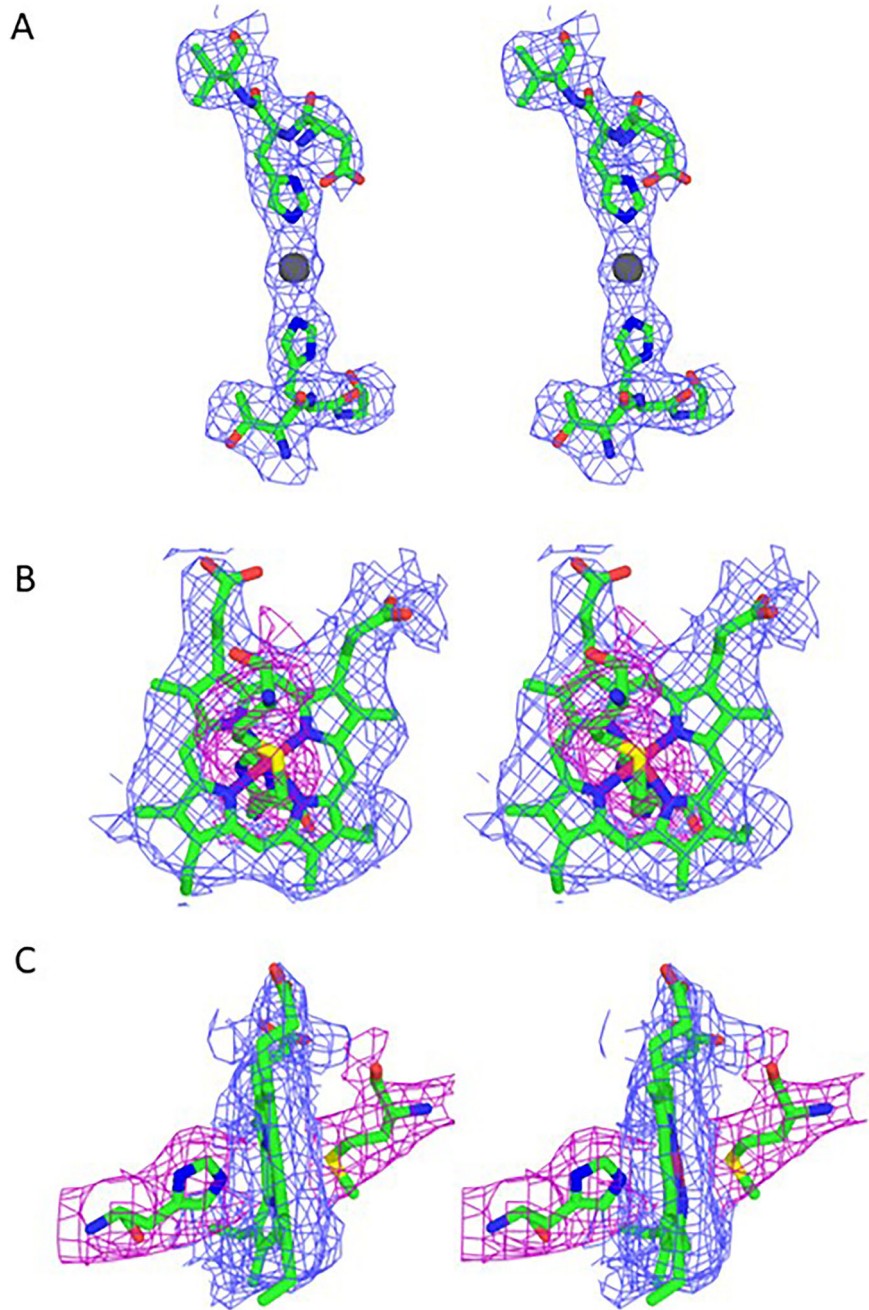

**Figure EV5. Potential density maps.**

(A) Stereo figure showing the potential density for Na1 and the coordinating histidine residues. Asp1208-His1209-Val1210 are positioned above Na1 while Gly537-His538-Thr539 are below. (B) Stereo figure showing the potential density for Met79, His254 and the heme group modeled at the interface of the head and neck domains. The isonet contoured around Met79 and His254 is in magenta, while that for the heme group is in blue. (C) As in (B), but rotated 90 degrees about the vertical axis to view the heme group edge on.

