## [Peer Review File · The EMBO Journal]

Commensal gut bacteria employ de-chelatase HmuS to harvest iron from heme

Arnab Kumar Nath, Ronivaldo da Silva, Colin Gauvin, Emmanuel Akpoto, Mensur Dlakic, C. Lawrence, and Jennifer DuBois

Corresponding authors: Jennifer DuBois (jennifer.dubois1@montana.edu) , C. Lawrence (c.martin.lawrence@gmail.com)

Review Timeline:

Submission Date:	11th Dec 24
Editorial Decision:	3rd Feb 25
Revision Received:	5th Jul 25
Editorial Decision:	2nd Aug 25
Revision Received:	8th Aug 25
Accepted:	22nd Aug 25

Editor: Ieva Gailite

Transaction Report:

Dear Dr. DuBois,

Thank you for submitting your manuscript for consideration by the EMBO Journal. We have now received comments from a full set of reviewers, which are included below for your information.

As you can see, while the referees per se find the study of interest, they also indicate that further information on the structural data would be needed (reviewers #1 and #3). Furthermore, referee #3 requests structure-based mutational follow-up analysis in order to strengthen the findings, and reviewers #2 and #3 find that further support to the proposed HmuS enzymatic activity would be needed. Finally, reviewer #2 in their point 1 raises the question on whether the HmuS activity can be observed in a microbiota setting, indicating that protoporphyrin levels are low in the feces.

Based on the overall interest expressed in the reports of both reviewers and your input during the pre-decision consultation, I would like to invite you to revise the manuscript in response to the reviewers' comments. I appreciate that it is not feasible to experimentally address point 1 by reviewer #2 within a single revision round. However, please add a discussion on the likely fate of microbiota-produced protoporphyrin. I should add that it is The EMBO Journal policy to allow only a single major round of revision and that it is therefore important to resolve the main concerns at this stage.

We generally allow three months as standard revision time. Should you foresee a problem in meeting this deadline, please let me know in advance to discuss an extension. As a matter of policy, competing manuscripts published during this period will not negatively impact on our assessment of the conceptual advance presented by your study. However, please contact me as soon as possible upon publication of any related work to discuss the appropriate course of action.

When preparing your letter of response to the referees' comments, please bear in mind that this will form part of the Review Process File and will therefore be available online to the community. For more details on our Transparent Editorial Process, please visit our website: <https://www.embopress.org/page/journal/14602075/authorguide#transparentprocess>. Please also see the attached instructions for further guidelines on preparation of the revised manuscript.

Please feel free to contact me if you have any further questions regarding the revision. Thank you for the opportunity to consider your work for publication. I look forward to receiving your revised manuscript.

With best regards,

Ieva

- a point-by-point response to the referees' comments, with a detailed description of the changes made (as a word file).
- a word file of the manuscript text.

- individual production quality figure files (one file per figure)
 - a complete author checklist, which you can download from our author guidelines (<https://www.embopress.org/page/journal/14602075/authorguide>).
 - Expanded View files (replacing Supplementary Information)
- Please see out instructions to authors
<https://www.embopress.org/page/journal/14602075/authorguide#expandedview>
- a Reagents and Tools Table as part of the Methods section, which can be downloaded from our author guidelines (<https://www.embopress.org/page/journal/14602075/authorguide#structuredmethods>)

We realize that it is difficult to revise to a specific deadline. In the interest of protecting the conceptual advance provided by the work, we recommend a revision within 3 months (4th May 2025). Please discuss the revision progress ahead of this time with the editor if you require more time to complete the revisions.

Referee #1:

How iron from dietary heme is utilised in the gut remains largely unknown. In this paper, Nath et al. characterized an enzyme, HmuS, in anaerobic bacteria in the gastrointestinal tract that is responsible for the anaerobic breakdown of heme into protoporphyrin IX and iron. HmuS is a membrane-bound protein and part of a 6-gene operon. The authors showed it to be NADH-dependent and inhibited by ATP. A cryo-EM structure indicates that HmuS has homology to families of metal chelataes. Based on the observation of a bound heme cofactor and two cation positions, together with a bioinformatic analysis of conserved features Nath et al. propose a tentative model for the mechanism.

This is an interesting paper that sheds light on an apparently ubiquitous but so far unknown mechanism of dealing with iron in the human gut. It is well-written and the conclusions are plausible. My only concern is that the structural work is not clearly presented. There is no figure showing the EM density. A global resolution estimate is not a good measure for the quality of a map.

Specific points that need attention:

The structure of HmuS was determined by cryo-EM. However, EM data is not shown in the paper, and details about the structure determination are scarce. At the very least, some figures should be added to the supplementary material (see below). Why are some images taken at 15 degrees tilt? Was there a preferred orientation problem? This is not addressed in the paper. A plot of viewing directions should be added to figure S12 to show that the dataset was sufficiently isotropic.

The processing procedure yields a single structure. Is this really the case? Is for example the heme position fully occupied? The connection between the head and neck is considered to be a "hinge", which would suggest flexibility, resulting in local resolution decrease. EM density should be shown for some important regions, like the bound cations and the heme. On p. 10 it is stated that there are poorly structured regions, possibly due to proteolysis, but this is never mentioned again. A figure colored by local resolution should be added in the supplement.

Figure S12 needs to be improved:

The processing scheme is inconsistent. It starts with results in boxes and procedures along arrows, which makes sense. But then steps are skipped (particle picking) and micrographs classified. Images of 2D and 3D classes are missing. Next the procedures are in boxes and results along the arrows. This whole scheme doesn't add anything to the more detailed description in the Methods.

Further, the colour bar in the bottom right corner does not serve any purpose.

A map coloured by local resolution should be added (which is much more informative than the histogram in the bottom left corner, which can be removed).

Minor issues:

In the abstract the authors state that "The proposed structure-based mechanism for iron removal by HmuS links biosynthetic and biodegradative pathways for haem, chlorophyll, and vitamin B12." This suggests a biochemical conversion of these compounds,

but I think what is meant is that enzymes in these pathways share a mechanism. This should be rephrased.

Cryo-EM methods:

- "heterogenous/homogenous refinement" should be "heterogeneous/homogeneous refinement".
- "Fourier shell coefficient" should be "Fourier shell correlation".
- Reference numbers in this section (60-64) appear to be mixed up. Please check.
- Scheme is misspelled as schaeem.
- The processing scheme is in figure S12, not S14.
- "5,929,8976 particles" has a digit too many.

Figure S17: The residue numbers for HmuS on the figure do not match the conserved residues mentioned in the text (and in the structure file). There appears to be an offset of 29 residues. The correct numbering should be used on the figure.

Table S3:

The final number of particles has a digit too many.

FSC 0.143 (masked/unmasked) should be (unmasked/masked).

Model vs. data FSC has to be stated at FSC 0.5, not 0.143. This latter criterion can only be used for data split in two independent halves.

Referee #2:

The manuscript by Nath et al reports on details and structure of HmuS and provides physiological, biochemical, and structural evidence for the anaerobic removal of iron from heme. They show that HmuS is a membrane-bound, NADH-dependent de-chelataase that removed Fe from heme leaving an intact porphyrin macrocycle. This is an extension of this groups previously published work (Mesle et al., 2023, ref #12 in the current manuscript).

Overall the data are well presented and support their conclusions. The figures and legends are well done and the methods are described in considerable detail. My only comment that is less than enthusiastic is that the discussion/results about potential roles of some residues in HmuS seems to be a bit of a reach given how little is actually known about how this enzyme functions. Perhaps some of those discussions/comments are more appropriate for a manuscript that describes the impact of site directed mutagenesis on HmuS. Secondly, I found the final sentence of the Conclusions (...could play heretofore unimagined roles in the cellular biology of chlorophyll, vitamin B12/cobalamine, haem, and their constituent metals) to be a bit over the top.

- 1) The question that comes to mind is why is there no protoporphyrin in feces since these bugs are dechelating heme and accumulating PPIX ? What is happening to it? It would be interesting to do some labeling/tracer experiments to see what breakdown products may occur and then identify the mechanism for this action. I'm not aware of any biological process that breaks down free PPIX.
- 2) Is the gut flora of rodents, who acquire Fe more from "free iron" rather than heme, different from humans who prefer heme iron?
- 3) There is an implication that mucosal uptake of iron is as free iron liberated by the action of the dechelataase rather than by active uptake of intact heme (pg 21). This would imply that these gut organisms are dechelating heme and thereby providing iron to the host. This doesn't make sense since the reaction is energy consuming and generates PPIX to make and release the free iron without a clear return to them. It would seem more reasonable that these organisms would finely regulate the dechelataase reaction to diminish the amount of PPIX being produced and accumulated.
- 4) On pg 21-22 there is an implication that the basis for porphyrin accumulation in cancer cells (in PDT) may result from the PPIX released by HmuS action. This is not accurate. The basis for accumulation of free PPIX in PDT treatment of tumor cells is well researched and known to be cellular derived. In some necrotic tissue that has bacterial infiltration one may see PPIX fluorescence, but this will be microbe associated porphyrin, not porphyrin transferred to cancerous cells.

Referee #3:

The manuscript by Nath et al describes the first de-chelataase in the Bacteroidetes. The physiological, biochemical and structural approach is fairly rigorous and detailed. The authors performed biochemical analysis of the protein in terms of the required cofactors and enzyme activity in the anaerobic removal of iron from heme. Based on the structural analysis by cryo-EM the authors propose a mechanism for the enzyme by comparing and contrasting the structural features of HmuS to the related chelataases CobN and ChlH. Overall, the work is interesting and expands on the diversity of heme catabolism in bacteria and the importance of the physiological environment in which these organisms must compete for limited micronutrients. However, I have some concerns that detract from the significance of the research. These would need to be addressed to absolutely confirm HmuS is indeed a dechelataase.

1. In Figure 2 the authors show the restriction of Fe²⁺ (BPS) leads to increased production of PPIX as a result of heme utilization. However, is it possible that restriction of Fe²⁺ leads to increased PPIX from endogenous sources? Perhaps a future experiment with ¹³C-labeled heme or Fe would confirm this. In any of the experiments were the intracellular iron levels

determined by ICP-MS.

2. In Fig 2D following passage to reduce the heme carry over even the wild type was unable to grow in the absence of heme. Why would this be the case given the fact that Fe is required for growth based on Fig 2a?
3. In identifying a potential reductase the authors mention the 20-30kDa protein that is fractionated during purification. Do the authors attempt to add back the fraction to see if the NADH-dependent activity is restored? In the absence of heme or NADH do the authors extract PPIX the control reaction extractions would provide support for the in vitro activity.
4. In Table S3 (cryo-EM data collection, particle refinement, model fitting, and refinement) are all parameters to evaluate the structure. However, it would be helpful to show the local resolution, which can vary widely across electron potential maps generated by cryo-EM. The authors do state "global resolution of 2.6" but this does not help with understanding the variation in the local resolution. Similarly, only select views of the structure are shown, making it impossible to assess whether there is anisotropy (e.g. 3D-FSC). It would also be helpful for rigor to include the PDB validation report to indicate if the Q-score independently agrees with the global resolution reported by the authors.
5. On p. 17 the authors state "Critically, we also identified a bound haem at this interface that copurified with HmuS, with the iron coordinated by the strictly conserved Met-79 and His-254 residues (Figure 6). Given the structural similarity of the head and neck domain to class II chelatases and periplasmic binding proteins (above), there has been speculation that class I chelatases might utilize this site to coordinate ligands. However, this is the first time a tetrapyrrole has been found at this site" . This is a significant finding and key to the mechanism proposed would be strengthened by showing the electron potential map of the heme to confirm this.
6. The authors propose "The substrate haem could easily fit in the vicinity of Na⁺(1), which has conserved residues suggestive of heme ligation, including proximal His-1209/Asp-1208 and distal His-538/Glu-577 histidine-carboxylate dyads that could act as axial ligands or acid-bases, Ser/Gln residues and main chain amines at the N-terminus of helix V- α 3 that could interact with propionates, and a surface of conserved side chains lining the pocket". While this may be the case there is no real evidence of this. The manuscript would have been strengthened with some site-directed mutagenesis to support the proposed role of this region.

Overall the authors present a plausible model but this does not appear at this atage to have been rigorously tested and as such seems to be an incomplete story. More solid evidence of extra cellular heme leading to PPIX would strengthen this as well as more controls in the measurement of the in vitro activity.

Minor points

1. On page 10 Fig 3b should be Fig S8E for the SDS-PAGE showing the 20-30 kDa band?
2. In the methods the authors use heme and PPIX to generate standard curves for quantification purposes. How is the extraction efficiency determined is there an internal standard? If not, how many replicates are performed this is not stated in the methods or figure legends? It is difficult to see in most of the bar graphs the error. It would be preferable to plot the bar charts as box plots with the actual data points.

Note that our responses are in blue.

Referee #1 (Report for Author)

How iron from dietary heme is utilised in the gut remains largely unknown. In this paper, Nath et al. characterized an enzyme, HmuS, in anaerobic bacteria in the gastrointestinal tract that is responsible for the anaerobic breakdown of heme into protoporphyrin IX and iron. HmuS is a membrane-bound protein and part of a 6-gene operon. The authors showed it to be NADH-dependent and inhibited by ATP. A cryo-EM structure indicates that HmuS has homology to families of metal chelatas. Based on the observation of a bound heme cofactor and two cation positions, together with a bioinformatic analysis of conserved features Nath et al. propose a tentative model for the mechanism.

This is an interesting paper that sheds light on an apparently ubiquitous but so far unknown mechanism of dealing with iron in the human gut. It is well-written and the conclusions are plausible. My only concern is that the structural work is not clearly presented. There is no figure showing the EM density. A global resolution estimate is not a good measure for the quality of a map.

We thank the reviewer for carefully reading the paper and are gratified that we were able to convey some of the excitement of the results. We are glad for the opportunity to improve the presentation of the structural results.

Specific points that need attention:

The structure of HmuS was determined by cryo-EM. However, EM data is not shown in the paper, and details about the structure determination are scarce. At the very least, some figures should be added to the supplementary material (see below). Why are some images taken at 15 degrees tilt? Was there a preferred orientation problem? This is not addressed in the paper. A plot of viewing directions should be added to figure S12 to show that the dataset was sufficiently isotropic.

We thank the reviewer for the invitation to elaborate on data collection and processing. And yes, there was a slight orientation bias. We have added the following to the end of the second paragraph of the Methods for Cryo-EM single particle analysis:

“Data collection and initial processing were done with CryoSPARC Live. The initial viewing direction distribution showed a slight preferred orientation problem. To overcome this, we increased the number of 0° tilt angle exposures to 7,800 and then collected an additional 8,194 exposures with a 15° tilt angle.”

We've also added a final paragraph and an additional supplemental figure (S11) to the methods on orientation diagnostics. Note that in response to another query from this reviewer (below) we have now used focused 3D classification to resolve one structure with an ordered head domain, and a second structure in which the head domain is disordered. Thus this new paragraph on orientation diagnostics now describes results for two data sets.

“CryoSPARC orientation diagnostics (Figure S11) reported a broad distribution of azimuthal tilt angles for each data set. Thus, the minor preferred orientation problem noted during initial data collection was successfully overcome with the increased number of micrographs collected at two different tilt angles (0°, 15°). Consistent with this, the 3D FSC analysis⁶² gave a conical FSC Area Ratio (cFAR) of 0.68 and minimum and maximum directional resolutions of 2.84 and 2.53 Å for the ordered domain 1 structure. Similarly, the disordered domain 1 structure had a cFAR of 0.72 with minimum and maximum resolutions of 2.45 and 2.76 Å. CryoSPARC documentation suggests a

cFAR value above 0.5 serves as a reasonable threshold for the lack of preferred orientation. In addition, the Fourier sampling analysis^{63,64} implemented in cryoSPARC yielded Sampling Compensation Factors (SCF) of 0.891 and 0.864 for the respective ordered and disordered structures, where values above 0.81 generally indicate good, though not necessarily isotropic signal content.”

Supplemental figure S11 the traditional cryoSPARC Viewing Direction Distribution plot for each of the two data sets, as well as additional output from the cryoSPARC orientation diagnostics job, including i) the relative signal as a function of viewing angle, ii) Fourier sampling analysis and Sampling Compensation Factor, and iii) the 3D or conical FSC, showing cFAR for HmuS is 0.75. CryoSPARC documentation suggests a cFAR of 0.5 “serves as a reasonable threshold for the presence, or lack thereof, of preferred orientation”.

The processing procedure yields a single structure. Is this really the case? Is for example the heme position fully occupied?

We thank the reviewer for pushing us to answer this question. Using 3D classification focused on the head domain, we were indeed able to discern one structure with an ordered head domain, and a second structure with a disordered head domain. The heme refines to > 90% occupancy in the structure with the ordered head domain, indicating it is at or near full occupancy. However, density is lacking for heme in the structure with the disordered domain 1. The penultimate paragraph in the results now describes this result:

“In this context, density for the head domain in the initial single particle map was relatively well ordered, but significantly weaker than the rest of the structure. This prompted 3D classification efforts focused on domain 1 that resolved two distinct particle sets, one giving a map with strong density for the head domain (PDB 9D26), and a second in which the head domain is disordered (PDB ID 9P4S). Notably, the ordered structure retains strong density for the heme group at the interface of the head and neck domains. In contrast, heme density is absent in the disordered structure. These two structural snapshots are consistent with a mobile head domain in the absence of heme, that adopts an ordered conformation with heme is bound at the subunit interface.”

We have also inserted the following paragraph into the methods section for the single particle analysis:

“Subsequent efforts to resolve the MRI with focused 3D classification/refinement were unsuccessful. In contrast, a 3D classification job focused on domain 1 with the number of classes set to 4, gave 3 classes with no or extremely poor density for domain 1, but a 4th class (2,133,842 particles) with strong, well-defined domain 1 density. The three classes lacking domain 1 density were then combined (3,795,134 particles), and each of the particle subsets (ordered domain 1, disordered domain 1) were subjected to an additional round of homogenous refinement. Both maps showed strong density for domains 2-6. The 2,133,842 particle set refined to a global resolution of 2.60 Å by GSFSC and again showed strong domain 1 density, while the 3,795,134 particle set refined to 2.55 Å and lacked domain 1 density.”

The connection between the head and neck is considered to be a "hinge", which would suggest flexibility, resulting in local resolution decrease.

The local resolution map does indeed show slightly lower resolution for the head domain. We were hopeful that focused refinement might improve head domain resolution, but there was not a significant difference. The decreased resolution of the head domain is now clear in the local resolution map for 9D26 in figure S10.

EM density should be shown for some important regions, like the bound cations and the heme.

Thank you for the suggestion, we added an additional supplemental Figure S19 to show wall eyed stereo views of the potential density isonet and corresponding model for the bound cations in the central cleft (2a) and the heme at the head neck interface (2b, 2c).

On p. 10 it is stated that there are poorly structured regions, possibly due to proteolysis, but this is never mentioned again. A figure colored by local resolution should be added in the supplement.

The protease hypothesis was formulated in the early days of our structural analysis, as a potential explanation for the lack of density in the MRI. However, subsequent focused 3D variability analysis was unable to identify a subset of particles with even a poorly ordered MRI. Our experience with 3D variability analysis on other projects suggests that if even half the particles suffered proteolysis in the MRI, the other half should have been easily sorted for the presence of the MRI, perhaps even into multiple conformations. Thus, it seems the MRI is intrinsically disordered, with missing potential density, regardless of whether it has been proteolytically cleaved. For this reason, we have removed the poorly structured regions comment on page 10, though we are still very upfront about the proteolysis problem itself.

As mentioned above, local resolution maps updated with a new color scheme are included in Figure S10.

Figure S12 needs to be improved:

The processing scheme is inconsistent. It starts with results in boxes and procedures along arrows, which makes sense. But then steps are skipped (particle picking) and micrographs classified. Images of 2D and 3D classes are missing. Next the procedures are in boxes and results along the arrows. This whole scheme doesn't add anything to the more detailed description in the Methods.

We thank the reviewer for these comments. We have updated Figure S10 to include a consistent scheme, 2D and 3D classes, though only a subset of the 2D classes are shown.

Further, the colour bar in the bottom right corner does not serve any purpose.

The color bar was the key for the local resolution map, indicating the correlation between color and local resolution. It has been updated with our new color scheme.

A map colored by local resolution should be added (which is much more informative than the histogram in the bottom left corner, which can be removed).

We've updated the local resolution map in Figure S10 with a new color scheme and removed the histogram.

Minor issues:

In the abstract the authors state that "The proposed structure-based mechanism for iron removal by HmuS links biosynthetic and biodegradative pathways for haem, chlorophyll, and vitamin B12." This suggests a biochemical conversion of these compounds, but I think what is meant is that enzymes in these pathways share a mechanism. This should be rephrased.

This has been made more exact, as follows:

"The proposed structure-based mechanism for iron removal by HmuS is chemically analogous to the chelatases in unrelated heme biosynthetic pathways and homologous enzymes in the

biosynthetic pathways for chlorophyll and vitamin B12."

Cryo-EM methods:

- "heterogenous/homogenous refinement" should be "heterogeneous/homogeneous refinement".
- "Fourier shell coefficient" should be "Fourier shell correlation".
- Reference numbers in this section (60-64) appear to be mixed up. Please check.
- Scheme is misspelled as schaem.
- The processing scheme is in figure S12, not S14.
- "5,929,8976 particles" has a digit too many.

We apologize for these errors and thank the reviewer for catching them. These changes have been made.

Figure S17: The residue numbers for HmuS on the figure do not match the conserved residues mentioned in the text (and in the structure file). There appears to be an offset of 29 residues. The correct numbering should be used on the figure.

We thank the reviewer for noticing the inconsistency in numbering. It was caused by the missing 29 residues in the final structure. All the numbers in the updated figure correspond to residue numbers in the text.

Table S3:

The final number of particles has a digit too many.

FSC 0.143 (masked/unmasked) should be (unmasked/masked).

Model vs. data FSC has to be stated at FSC 0.5, not 0.143. This latter criterion can only be used for data split in two independent halves.

We apologize for these errors and thank the reviewer for catching them. They have been corrected.

Referee #2 (Report for Author)

The manuscript by Nath et al reports on details and structure of HmuS and provides physiological, biochemical, and structural evidence for the anaerobic removal of iron from heme. They show that HmuS is a membrane-bound, NADH-dependent de-chelatase that removed Fe from heme leaving an intact porphyrin macrocycle. This is an extension of this groups previously published work (Mesle et al., 2023, ref #12 in the current manuscript).

Overall the data are well presented and support their conclusions. The figures and legends are well done and the methods are described in considerable detail. My only comment that is less than enthusiastic is that the discussion/results about potential roles of some residues in HmuS seems to be a bit of a reach given how little is actually known about how this enzyme functions. Perhaps some of those discussions/comments are more appropriate for a manuscript that describes the impact of site directed mutagenesis on HumS.

The intended purpose of this paper is, as the reviewer has stated, to validate the enzyme's proposed function both in vitro and in the native bacterial host. We agree that in-depth investigations of structure-activity relationships must proceed in subsequent papers, probably several of them, that directly address questions about mechanism. We have also thought carefully about what needs to be included in this paper to help the reader make sense of the structural data it presents, and in light of the reported activity.

First, we described 3 reactions involving the enzyme: equilibrium binding of ≥ 1 ferriheme ligands,

reduction of the “site 1” heme by NADH (which is stably reduced as the protein is expressed in air), and conversion of a heme to PPIX. The last reaction occurs only in the presence of NADH and a stoichiometric excess of heme.

Second, the structural data revealed two ligand binding sites: a heme bound between domains 1 and 2, which we identify with the heme present in the as-isolated protein (site 1). Additionally, two cations occupy well conserved sites within the main body of the structure (site 2). The observed ligands invite the question: why are there two, and which of the 3 reported reactions takes place in either site?

The data from the original manuscript, with some additional information, has been further clarified and “mapped” onto the structure’s two binding sites. Doing so makes sense of the structure without stretching the story too far toward mechanism. The revised Results section describes HmuS activity as follows:

- The as-isolated HmuS has a site partially occupied by heme (Fig 3a). Titrimetric addition of heme initially augments the UV/vis absorbance spectrum of this HmuS-heme complex (Fig 3c-3d).
- Using the HmuS-heme complex with less than 1 equivalent of heme bound, we showed that excess NADH reduced the ferriheme to ferrous heme over time. However, we observed no further formation of PPIX + Fe(II) by UV/vis or by chemical analysis (Fig 3e – 3f).
- As the titration proceeds above the K_d of the high-affinity heme site ($K_d = 6 \mu\text{M}$), an additional heme binding site ($K_d > 6 \mu\text{M}$) with unique spectral features is accessed (Fig 3c). However, overlap between the spectra for the high affinity HmuS-heme complex and additional HmuS-heme complex(es) prevents us from obtaining clear titrimetric data or component UV/vis spectra describing the individual binding sites.
- To access additional heme binding sites, we incubated HmuS with a stoichiometric excess of heme, then rapidly removed loosely bound heme by centrifuge filtration. This results in a multi-heme-loaded HmuS (Fig 3f – 3g).
- Adding excess NADH to the heme-loaded HmuS results in changes in the Soret and Q-band regions of the UV/vis absorbance spectrum over time. Analysis of heme conversion before/after addition of NADH indicates a little more than one turnover of heme to PPIX. This indicates that more than one equivalent of heme must be bound to HmuS to observe heme-PPIX conversion.
- We examined the wt HmuS for PPIX production under turnover conditions: excess hemin and NADH added to the enzyme, followed by incubation for 40 min at 25 °C. For wt HmuS under these conditions, we consistently observed approximately one turnover of heme to PPIX per HmuS (Fig 6). This indicates that HmuS converts heme to PPIX and Fe(II), though the reasons for limited catalytic efficiency are not yet explained by these data. We offer a few possibilities – for example, the need for a cofactor, or product inhibition by PPIX. However, we refrained from delving too deeply into reasons for limited turnovers, as the point of the paper - to demonstrate that HmuS converts heme to PPIX – has already been amply made, and to go further would require speculation.
- We identified H538 as a focal point of “site 2”, for several reasons cited in the text. The H538A mutant expressed with the same efficiency and characteristic chromatography as the wt HmuS. It retained the ability to bind heme into the high affinity site, with K_d effectively unchanged.

- Under the same conditions described for the wt HmuS above, the H538A HmuS did not convert heme to PPIX + Fe(II) (Figure 6). A list of factors including: the complete loss of heme-PPIX conversion activity in the H538A mutant; the retention of heme binding and reduction activities at site 1 in both the wt and H538A HmuS proteins; the identification of a large, highly conserved hydrophobic cleft capable of accommodating a porphyrin at site 2; and the observation of a product-like cation bound at H538 in the cryoEM structure (Fig 5) instead of heme, all suggest that “site 2” is important for iron removal from heme. The roles of the two ligand binding sites and of individual amino acids in the catalytic mechanism, however, remain to be discovered in future work.

Secondarily, I found the final sentence of the Conclusions (...could play heretofore unimagined roles in the cellular biology of chlorophyll, vitamin B12/cobalamine, haem, and their constituent metals) to be a bit over the top.

The reviewer’s point is well-taken. Our statement at the end of the paper was delivered without elaboration and consequently fell flat. After reading the reviewer’s comment, we realized that we never actually included an explanation of what we intended by that sweeping conclusion. The following statement has been inserted into the text with appropriate literature references, immediately after the description of the species distribution of the informatically-identified superfamily sequences:

“We were surprised to find so many members of the parent superfamily in taxonomically distinct groups, sometimes with multiple paralogs in the same species. We wondered whether proteins like HmuS might have biological roles outside the hmu pathway. Salvage pathways using chelatase/dechelatase processes might be important, for example, for repurposing the PPIX generated by heme catabolism in the microbiome environment. Cofactors are metabolically expensive to build. Alternatively, metalation/demetallation could permit activation and deactivation of cofactors on an as-needed basis. Chlorophyll, for example, is known to be reversibly demetallated and converted to pheophytin by protonation, or remetallated with Zn(II) or Cu(II) to stabilize it against heat. These reactions, which are known to be catalyzed by enzymes that are not homologous to HmuS, offer a mechanism for photosynthetic organisms to protect themselves or adjust their photosynthetic efficiency according to diurnal or seasonal cycles. Metalation/demetallation could likewise be a means by which microbiome species adjust their cellular heme concentrations in response to shifting conditions. Alternatively, demetallation/remetallation could be essential steps toward further functionalization of ingested heme b around its perimeter, funneling it toward more specialized roles. Finally, a large number of HmuS homologs were identified in diverse species of Archaea. Methanogenic Archaea are known for their use of F430, a Ni(II)-dependent cofactor constructed from uroporphyrinogen: a central metabolic precursor of siroheme, F430, and heme b in many of these species. It is unclear how Archaea might use HmuS/CobN/ChlH homologs, particularly since these species do not possess complete hmu operons for heme assimilation, nor do they have aerobic pathways for generating chlorophyll or cobalamin.”

We also replaced the statement in the Conclusions section with a toned down version:

“We propose that flexible metal chelation/de-chelation, using the widespread HmuS homologs identified informatically in this work, may play unknown roles in the metallobiochemistry of diverse species.”

1) The question that comes to mind is why is there no protoporphyrin in feces since these bugs are dechelating heme and accumulating PPIX ? What is happening to it? It would be interesting to do some labeling/tracer experiments to see what breakdown products may occur and then identify the mechanism for this action. I'm not aware of any biological process that breaks down free PPIX.

This is a terrific question. In fact, we've generated a variety of labeled heme species in order to address it in the context of our mouse model (Coe et al.) and bacterial cultures/cocultures.

We also do not know of any pathway for further metabolizing PPIX. In the statement above, we articulate one hypothesis that we are testing in bacterial co-culture experiments: that PPIX produced by organisms like *B. theta* can be recycled by other bacteria in the GI tract consortium. It may be remetallated to make heme using HmuS proteins running in reverse. Though they are not abundant in the GI tract, species that produce ferrochelatase could also potentially consume and metallate PPIX. In the presence of O₂ and heme oxygenase (from the host animal or in a facultative species like *E. coli*), heme can be fully metabolized. Alternatively, there could be undiscovered pathways for PPIX metabolism in microbiome multicultures. Amazingly, a microbial enzyme catalyzing bilirubin conversion to stercobilin was just discovered in 2024. There are many unsolved mysteries in the GI tract.

It is notable that, when PPIX is generated by pathogenic *Bacteroides* species outside the GI tract – for example, *B. fragilis* infections in anaerobic surgical wounds – it appears to accumulate to levels that are easy to detect by fluorescence. This suggests that a mechanism for PPIX clearance exists in the GI tract, but not necessarily elsewhere.

The coproporphyrin and PPIX generated by pathogenic *H. pylori* inside the GI tract provide an exception. In this case, porphyrins are associated with *H. pylori*'s endogenous heme biosynthetic pathways and not with heme demetallation. It's not clear why *H. pylori* accumulates "significant" quantities of porphyrins, though the coproporphyrin, with no known salvage or metabolic pathway (beyond conversion to heme by the Gram positive pathway), would be potentially toxic both to the human host and to bacterial competitors.

(Hamblin MR, Viveiros J, Yang C, Ahmadi A, Ganz RA, Tolkoﬀ MJ. Helicobacter pylori accumulates photoactive porphyrins and is killed by visible light. Antimicrob Agents Chemother. 2005 Jul;49(7):2822-7. doi: 10.1128/AAC.49.7.2822-2827.2005. PMID: 15980355; PMCID: PMC1168670.)

2) Is the gut flora of rodents, who acquire Fe more from "free iron" rather than heme, different from humans who prefer heme iron?

This is another great question. Among humans, the microbiomes of vegetarians are different from those who have more omnivorous diets. Mice, being natural vegetarians, would be expected to acquire more iron from plant sources. Plant sources are not devoid of heme but are certainly less enriched in it, relative to red meat.

Conventional mouse microbiota are rich in members of the Bacteroidetes phylum – which possess *hmu* operons – membership may differ at lower taxonomic levels. For example, where humans may have abundant members of the *Bacteroides* genus, in mice, one might find more *Prevotetella* sp. Physiologically, these two genera are very similar. They are each heme auxotrophs possessing *hmu* operons. Each has an abundance of carbohydrate metabolizing pathways.

To really address these issues, mice can be made germ-free and subsequently populated with defined microbiota, including natural human consortia, mouse consortia, or single strains (e.g., *B. theta*, or the *hmuS* knock out of *B. theta*). These are experiments that we hope to do in the future using our mouse model for anemia, and feeding it defined diets abundant in heme or non-heme iron..

3) There is an implication that mucosal uptake of iron is as free iron liberated by the action of the dechelatase rather than by active uptake of intact heme (pg 21). This would imply that these gut organisms are dechelating heme and thereby providing iron to the host. This doesn't make sense

since the reaction is energy consuming and generates PPIX to make and release the free iron without a clear return to them. It would seem more reasonable that these organisms would finely regulate the dechelatase reaction to diminish the amount of PPIX being produced and accumulated.

We regret that this is not the message we intended to convey. This portion of the text has now been clarified as described below.

On p. 21, we wrote:

“Whether heme is derived from the host diet or the recycled lining of the intestine, HmuS-mediated heme de-chelation presents what may be a major mechanism for host-microbe heme metabolism, particularly given the absence of an unequivocally defined heme transporter in human intestinal epithelium and the dependence of *Bacteroidetes* sp. on exogenously supplied heme for survival.”

Intestinal heme transporters – molecules that would mediate direct intake of dietary heme into human intestinal epithelia – have a challenging history. In 2005, Heme Carrier Protein 1 (HCP1) was initially identified as *an* or perhaps *the* intestinal heme importer. However, it was later demonstrated to serve as a pH-dependent folate transporter and renamed: “proton-coupled folate transporter.” Since that time, mechanisms of heme import into the intestine have remained challenging to define. Some evidence suggests that these mechanisms could be both species and tissue dependent. Recent work with the nematode *Caenorhabditis elegans* suggested the involvement of multiple, paralogous transmembrane Heme Responsive Gene (HRG) transporters in controlling heme uptake in that organism, with different paralogs acting in a tissue-specific manner. Transcript data suggest that HRGs may play a role in heme transport in multiple metazoans, including humans, though direct evidence that human HRG-1 is responsible for dietary heme import into the intestinal epithelial cells, to our knowledge, has not been published.

Regardless of how the intestinal heme transporter story ultimately turns out, we can be sure that dietary heme, as well as heme produced from recycling of the intestinal epithelium, interacts with the GI tract microbiome of humans. We also know that the GI microbiome is rich in heme auxotrophs. We have shown here that a common species of this kind grows well with heme as its sole source of iron. Under conditions where these species demetallate heme to meet their needs for non-heme iron – hypothetically, a diet rich in red meat – some of the liberated iron will ultimately become the property of the host, as the microbiome itself continuously turns over. In the absence of an unambiguous intestinal heme transporter, and in the presence of a microbial consortium with an obligate heme requirement and a mechanism for degrading heme, we conclude that the resident microbiome provides at least one means of supplying Fe(II) to the divalent metal transporter (DMT-1) in the human host. Importantly, we do not intend to imply that microbiome species undertake heme metabolism *for the purposes of feeding the host animal*. Rather, given the avidity of some of the constituent microbiome species for heme, and the constant turnover of the bacterial cells themselves, some of the bacterial heme and heme-liberated iron will be steadily relinquished to the host.

To clarify this message, we have changed the text above to read:

“Whether heme is derived from the host diet or the recycled lining of the intestine, HmuS-mediated heme de-chelation presents one mechanism for cooperative host-microbe heme metabolism. *Bacteroidetes* sp. depend on exogenously supplied heme for survival and grow robustly with heme as their source of iron, even at concentrations up to 0.5 mM. Some of this microbiome-associated iron will be steadily surrendered to the host as the bacterial cells die off and are resorbed. The significance of this iron pool to the host is unclear, though we note that a transporter for direct heme uptake across the human intestinal epithelium has been very challenging to identify.”

See for example:

Chen, C.; Hamza, I. Notes from the Underground: Heme Homeostasis in *C. elegans*. *Biomolecules* **2023**, *13*, 1149. <https://doi.org/10.3390/biom13071149>

Qiu A, Jansen M, Sakaris A, Min SH, Chattopadhyay S, et al. 2006. Identification of an intestinal folate transporter and the molecular basis for hereditary folate malabsorption. *Cell* **127**(5):917–28
Shayeghi M, Latunde-Dada GO, Oakhill JS, Laftah AH, Takeuchi K, et al. 2005. Identification of an intestinal heme transporter. *Cell* **122**(5):789–801

4) On pg 21-22 there is an implication that the basis for porphyrin accumulation in cancer cells (in PDT) may result from the PPIX released by HmuS action. This is not accurate. The basis for accumulation of free PPIX in PDT treatment of tumor cells is well researched and known to be cellular derived. In some necrotic tissue that has bacterial infiltration one may see PPIX fluorescence, but this will be microbe associated porphyrin, not porphyrin transferred to cancerous cells.

We thank the reviewer for noting this, and have stricken this part of the text to clarify that tumor-associated PPIX is not microbially derived.

Referee #3 (Report for Author)

The manuscript by Nath et al describes the first de-chelatase in the Bacteroidetes. The physiological, biochemical and structural approach is fairly rigorous and detailed. The authors performed biochemical analysis of the protein in terms of the required cofactors and enzyme activity in the anaerobic removal of iron from heme. Based on the structural analysis by cryo-EM the authors propose a mechanism for the enzyme by comparing and contrasting the structural features of HmuS to the related chelatases CobN and ChlH. Overall, the work is interesting and expands on the diversity of heme catabolism in bacteria and the importance of the physiological environment in which these organisms must compete for limited micronutrients. However, I have some concerns that detract from the significance of the research. These would need to be addressed to absolutely confirm HmuS is indeed a dechelatease.

1. In Figure 2 the authors show the restriction of Fe²⁺ (BPS) leads to increased production of PPIX as a result of heme utilization. However, is it possible that restriction of Fe²⁺ leads to increased PPIX from endogenous sources? Perhaps a future experiment with ¹³C-labeled heme or Fe would confirm this. In any of the experiments were the intracellular iron levels determined by ICP-MS.

Bacteroides thetaiotaomicron (*B. theta*) is a heme auxotrophic bacterium, meaning it requires exogenous heme for growth as it is unable to synthesize heme endogenously. This is generally true for members of its genus and class, and reflected in the lack of a heme biosynthesis operon in its genome. We demonstrated its dependence on exogenous heme by serially passaging *B. theta* into heme-free media. We observed that 5 passages were required to completely eliminate heme from the inoculum and stop growth.

Heme therefore serves as an essential growth factor that must be acquired from external sources. Bathophenanthroline disulfonic acid (BPS), a membrane-impermeable Fe(II) chelator, was employed to reduce the availability of non-heme iron in the minimal medium, thereby ensuring that heme was the sole accessible iron source. The same approach has been used in other studies requiring iron restriction.

We have measured cellular iron concentrations using ICP-MS under numerous growth conditions, as part of a proteomic/metalloproteomic study showing how *B. theta* compensates for limitations to its heme or non-heme iron. This work is now published as da Silva et al., *Frontiers*, 2025 (cited in

the text).

2. In Fig 2D following passage to reduce the heme carry over even the wild type was unable to grow in the absence of heme. Why would this be the case given the fact that Fe is required for growth based on Fig 2a?

Specifically, Figure 2c illustrates the growth of the wild-type strain and *hmuS* mutants in minimal medium (MM) supplemented with 15 μ M heme, either with BPS (red lines; heme serving as the sole iron source) or without BPS (purple lines; both heme and non-heme iron sources available), using a pre-inoculum containing heme (without serial passaging). Under these conditions, all strains grow when non-heme and heme iron are available (purple lines). However, only the wild-type strain grows under conditions of non-heme iron limitation (red lines), where heme is the sole iron source. This finding suggests that the HmuS protein is essential for iron extraction from heme in the wild-type strain. Iron sustains bacterial growth when heme is also present (purple lines).

Furthermore, when serial passaging is performed to prepare cells for inoculation, a critical observation is made. If heme is absent, none of the strains grow in the presence of Fe(II) alone (blue lines, Fig. 2d). However, when heme is added to the MM (without BPS), all strains from the same serial passaging pre-inoculum exhibit growth (purple lines, Fig. 2d). These findings demonstrate that heme is not only a vital iron source but also serves an essential cellular function, possibly as a cofactor for specific proteins, although this role remains to be fully elucidated.

In summary, these results strongly support the following conclusions: (i) iron is an essential metal for *B. theta* and can be acquired from heme by the wild-type strain, (ii) Hmu proteins play a crucial role in iron extraction from heme, and (iii) in the absence of heme, Fe(II) alone cannot support growth, highlighting the necessity of heme for *B. theta* survival.

3. In identifying a potential reductase the authors mention the 20-30kDa protein that is fractionated during purification. Do the authors attempt to add back the fraction to see if the NADH-dependent activity is restored?

We attempted these experiments. However, these depend on isolating a small amount of an unknown, endogenous *E. coli* protein with a potential moonlighting role toward HmuS. As a consequence, we could not carry out the experiments in a quantitative way.

At the same time, the goals of this manuscript are to demonstrate that HmuS converts heme to PPIX, and to establish that reaction within the context of the novel HmuS structure. Rather than look for added factors or conditions that could potentially enhance HmuS's efficiency, we have instead focused on presenting the most rigorous description of HmuS itself.

We have consequently pared back the discussion of a possible moonlighting *E. coli* reductase in the less-pure fractions of recombinant HmuS. At this point, it is a speculative discussion that leads away from the major question posed by the paper.

At the same time, we included additional data in a reorganized Figure 3, described extensively in the response to Reviewer 2 above. These results showed directly and unequivocally that multi-heme-loaded HmuS converts heme to PPIX in an NADH-dependent manner.

In the absence of heme or NADH do the authors extract PPIX the control reaction extractions would provide support for the in vitro activity.

These controls were carried out previously but were not explicitly shown or emphasized in the text. They have been added to the revised figure 6.

As described in the response to Reviewer 2, we show that binding ≤ 1 equivalent of heme to HmuS, followed by addition of NADH, is sufficient for reducing the heme, but does not result in further UV/visible absorbance changes consistent with PPIX production. Extracted reactions and fluorescence data do not indicate the presence of PPIX.

We then equilibrated HmuS with a small excess of heme and removed loosely bound heme by rapid buffer exchange. Addition of NADH to this pre-formed complex resulted in a new set of UV/visible absorbance changes over time. Extraction and analysis of the complexes, once the UV/visible changes were complete, identified turnover of roughly half of the initially bound heme to PPIX.

In the new figure 6, we added no-heme, no-NADH, and no-protein reaction controls to reactivity data measured under multiple-turnover conditions. Figure 6 also documents a strict requirement for H538 (the proposed site of substrate binding) for heme conversion to PPIX.

4. In Table S3 (cryo-EM data collection, particle refinement, model fitting, and refinement) are all parameters to evaluate the structure. However, it would be helpful to show the local resolution, which can vary widely across electron potential maps generated by cryo-EM. The authors do state "global resolution of 2.6" but this does not help with understanding the variation in the local resolution. Similarly, only select views of the structure are shown, making it impossible to assess whether there is anisotropy (e.g. 3D-FSC). It would also be helpful for rigor to include the PDB validation report to indicate if the Q-score independently agrees with the global resolution reported by the authors.

The local resolution maps are included in Figure S10 for both the ordered (9D26) and disordered (9D4S) structures that were resolved by focused 3D classification (see response to reviewer 1 above). In addition, orientation diagnostics that include Fourier sampling analysis^{63,64} are now presented in Figure S11. The Sampling Compensation Factors (SCF) were 0.891 and 0.864 for 9D26 and 9D4S, respectively. According to the cryosparc documentation, values above 0.81 generally indicate good, though not necessarily isotropic signal content. Finally, the PDB validation reports have been uploaded with the submission. Please contact the editor if they are not visible this time.

5. On p. 17 the authors state "Critically, we also identified a bound haem at this interface that copurified with HmuS, with the iron coordinated by the strictly conserved Met-79 and His-254 residues (Figure 6). Given the structural similarity of the head and neck domain to class II chelatasases and periplasmic binding proteins (above), there has been speculation that class I chelatasases might utilize this site to coordinate ligands. However, this is the first time a tetrapyrrole has been found at this site". This is a significant finding and key to the mechanism proposed would be strengthened by showing the electron potential map of the heme to confirm this.

Thank you for the suggestion, we added an additional supplemental Figure S19 to show wall eyed stereo views of the potential density isonet and corresponding model for the bound cations in the central cleft (2a) and the heme at the head-neck interface (2b, 2c).

6. The authors propose "The substrate haem could easily fit in the vicinity of Na⁺(1), which has conserved residues suggestive of heme ligation, including proximal His-1209/Asp-1208 and distal His-538/Glu-577 histidine-carboxylate dyads that could act as axial ligands or acid-bases, Ser/Gln residues and main chain amines at the N-terminus of helix V- α 3 that could interact with propionates, and a surface of conserved side chains lining the pocket". While this may be the case there is no real evidence of this. The manuscript would have been strengthened with some site-directed mutagenesis to support the proposed role of this region.

We expect that extensive mechanism-focused studies will be necessary to address the many relevant structure-activity questions that we are asking about HmuS. These studies will necessarily require multiple mutations and spectroscopic approaches, and possibly additional CryoEM structures. In short: thoroughly testing a proposed mechanism is a long-term aim. However, in this paper, our focus is well-defined: to demonstrate the novel enzymatic activity of HmuS.

With that in mind, of the many possible experiments that we could potentially do, we used mutagenesis to test our hypothesis that “site 2,” defined by the strictly conserved H538 ligand to the sodium cation, is involved in heme binding and is necessary for heme/PPIX turnover. This residue is strictly conserved superfamily-wide. It is surrounded by a cavity that easily accommodates a modeled protoporphyrin IX ligand. The cavity itself is lined with strictly conserved hydrophobic residues extending toward a region of high polarity, evocative of a binding site for the propionate side chains of heme. Because the unusual cis configuration of H538 is conserved in the available structure of cobaltochelatase, this seemed to be a residue of singular significance for the HmuS reaction. To test this, we generated and examined the H538A mutant, noting four key results:

1. H538A HmuS was purified with yields comparable to wild type HmuS, and analytical size exclusion chromatography (SEC) supported indicated identical apparent molecular weights. These observations support the conclusion that H538A HmuS is intact and folded.
2. The H538A HmuS, as-expressed and as-purified, had UV/vis spectra indicative of substoichiometric heme bound in an apparently homogenous site, identical to that of the wt protein. UV/vis spectra of heme proteins are strongly influenced by the identities of the heme-iron coordinating ligands. Since the as-purified wt protein was shown via cryoEM to bind heme in site 1, we concluded that the mutant protein retains the ability to bind heme in site 1 (Fig 6A).
3. The H538A site 1 binds heme with an unchanged K_d, relative to wt.
4. The H538A HmuS was assayed for heme dechelation activity under the same conditions that resulted in PPIX production by the wt HmuS. These data showed that no PPIX was formed by the H538A mutant. The experiment was repeated 6 times using two independent preparations of the mutant protein.

Overall the authors present a plausible model but this does not appear at this stage to have been rigorously tested and as such seems to be an incomplete story. More solid evidence of extra cellular heme leading to PPIX would strengthen this as well as more controls in the measurement of the in vitro activity.

We have explicitly added several controls into the text/SI to strengthen the presentation of the data. We also point out that *B. theta* is incapable of producing heme endogenously (see above). The only way it can make PPIX is via demetallation of heme, a reaction that we have validated by several means under a variety of conditions:

We have provided rigorous physiological (growth), analytical, and genetic evidence of heme uptake and conversion to PPIX (and iron) by *B. theta* in vivo, including interrogation of two separate *B. theta* strains bearing *hmuS* transposon insertions. Production of PPIX increases in proportion to the concentration of the BPS chelator (Mesle et al., 2024). This induces a greater demand for non-heme iron which must be derived from heme. Because *B. theta* is incapable of producing heme endogenously, the only possible source of PPIX in these experiments is the exogenously provided heme, which we have shown in numerous control experiments is not spontaneously demetallated under conditions used in our studies, either in cellulo or during analyses. In short: we have provided an exceedingly rigorous demonstration of heme-to-PPIX conversion by whole *B. theta* cells, and shown that this reaction depends on a functional *hmuS* gene.

The *in vitro* activity has similarly been rigorously demonstrated under limited turnover conditions from preformed HmuS-heme complexes, as well as under catalytic conditions using HmuS and excess heme/NADH. The catalytic reaction is inefficient but nonetheless documented reproducibly with numerous repetitions and controls. We note that heme oxygenase is likewise inefficient due to product inhibition. In that case, biliverdin reductase alleviates inhibition by facilitating removal of the highly hydrophobic product. We hypothesize that HmuS may experience product inhibition, benefiting from a chaperone to retrieve PPIX from the protein. Future work will address hypotheses about what controls the efficiency of the HmuS reaction, as well as how the protein's structure mediates its function. For now, we believe we have presented a long and detailed study that meets and exceeds the necessary bar for demonstrating this enzyme's function.

Minor points

1. On page 10 Fig 3b should be Fig S8E for the SDS-PAGE showing the 20-30 kDa band?

The discussion of a possible reductase has been removed from the text as overly speculative and beyond the intended scope of the paper, as described above.

2. In the methods the authors use heme and PPIX to generate standard curves for quantification purposes. How is the extraction efficiency determined is there an internal standard?

As described on pg 31, to quantify heme and PPIX by HPLC we used a standard curve performed using hemin chloride and PPIX.

“High-performance liquid chromatographic (HPLC) analysis of heme and protoporphyrin IX

Chromatography was carried out using a linear gradient with solution A (ultrapure water + 0.1% trifluoroacetic acid, TFA) and solution B (ACN + 0.1% TFA), flow rate of 1 mL min⁻¹, oven temperature at 25 °C (Figure S1). 30 µL samples were injected onto the column. Standard curves of hemin and PPIX (0.1, 0.5, 1, 5, 10, 50 µM) were prepared to quantify the extracted heme and PPIX via peak integration.”

For the HmuS activity assay, control mixtures were employed, matching the concentration and incubation time of the reaction mixtures, to assess whether heme would be converted into PPIX in the absence of HmuS or *B. theta* lysate fractions. The control mixture consisted of NADH and/or ATP (depending on their presence in the activity assay), combined with hemin and the extraction solvent. A blank was also performed to verify that hemin was not contaminated with PPIX, consisting of hemin and the extraction solvent. This control served as a negative control to calculate the net PPIX production, which was determined by subtracting the negative control value from the reaction value (reaction - negative control = produced PPIX). Experiments were carried out in triplicate. See sections **“Heme conversion to PPIX”** on pg 31, and **“Monitoring heme turnover to PPIX by purified HmuS”** on pg 33.

These methods and a discussion of the extraction efficiencies of several solvents (for heme, PPIX, and the two in combination) are described in Figure S1 and are a focus of a recent *Methods in Enzymology* paper by our group (cited above and in the text).

If not, how many replicates are performed this is not stated in the methods or figure legends? It is difficult to see in most of the bar graphs the error.

These changes have been made. The number of replicates is shown in Figs. 2 and 3. “Triplicate results were averaged. Error bars represent standard deviation.” While this is true of the data presented in the manuscript, at this point – all of the experiments in the manuscript have been repeated numerous times.

It would be preferable to plot the bar charts as box plots with the actual data points.

We acknowledge and concur with the reviewer's perspective; a boxplot effectively illustrates data

dispersion, the median, minimum and maximum values, as well as potential outliers. Our experiments were conducted in triplicate, yielding consistent replicate values, as evidenced by the low standard deviation and the absence of outliers. Consequently, the resulting boxplot exhibits a compact interquartile range, with minimal separation between the minimum and maximum values. This, in turn, complicates both the differentiation of experimental groups by color and the clear visualization of the median line.

Furthermore, considering that Figure 2 comprises six panels, we aimed to emphasize various key elements within the graphs to enhance result interpretation (including wild-type, *hmuS* mutants, presence or absence of BPS, cell membrane, PPIX, heme, and serial passages). We determined that a bar chart was the most appropriate choice for representing the data, as it allows for clear differentiation using color as well as solid and striped bars. The bar graphs effectively depict these results without compromising their interpretability and provide an accurate representation of heme and PPIX values, along with their respective standard deviations.

Dear Dr. DuBois,

Thank you for submitting a revised version of your manuscript. We have now received input from all original reviewers, who are satisfied with the revisions and now recommend acceptance of the manuscript.

There now remain only a few editorial points that need to be addressed before I can extend official acceptance of the manuscript:

1. Please make sure that the order of the sections in the manuscript is as follows: abstract, introduction, results, discussion, materials & methods, data availability section, acknowledgments, disclosure statement and competing interests, references, main figure legends, tables, expanded figure legends
2. Please add all information listed in the "Comments" box in our online submission system to the lines with the funding information - there is no need for the system to validate the entries at this point, but the list must be complete.
3. Please remove the list of Supplemental Information from the manuscript text.
4. In the "Methods" section, please add a disclaimer for BioRender use following the following format:
Graphics:
(some of the... OR Figure #... OR synopsis) Graphics were created with BioRender.com.
5. In the "Data Availability" section, please remove the sections with the headings "Resource Availability", "Lead Contact" and "Materials Availability". Please add add resolvable links to the listed datasets. More information about the format of this section can be found here: <https://www.embopress.org/page/journal/14602075/authorguide#dataavailability>.
6. CRediT has replaced the traditional author contributions section because it offers a systematic, machine-readable author contributions format that allows for more effective research assessment. Please remove the Authors Contributions from the manuscript and use the free text boxes beneath each contributing author's name in our online submission system to add specific details on the author's contribution. More information is available in our guide to authors.
7. Please rename 'Competing interest statement' section into 'Disclosure and competing interests statement' (further info: <https://www.embopress.org/page/journal/14602075/authorguide#conflictsofinterest>).
8. Please update references according to The EMBO Journal style - it should be alphabetical. Please also remove the DOIs. Please see further information here: <https://www.embopress.org/page/journal/14602075/authorguide#referencesformat>
9. In the Appendix, please add page numbers to the table of contents and correct the nomenclature to "Appendix Figure S1" etc. and "Appendix Table S1" throughout.
10. Please rename Documents S2-S5 into Dataset EV1-EV4.
11. Figure panels 1A-B; 4A-F; 5A-D; 6B are not mentioned in the manuscript text. Please add the corresponding callouts.
12. Please make sure that the figures are all called out in sequential order - currently, Fig 6A is called out before Fig 5E.
13. There are references to "data not shown" on pages 5, 7 and 8. According to our policy, which does not permit references to "data not shown", please include this information in the Appendix. Please see also <https://www.embopress.org/page/journal/14602075/authorguide#unpublisheddata>.
14. In our standard image check, we noticed that the Western blot image has been reused in Figure 3B, 6B and Appendix Fig S8A. If this is intentional, please clearly indicate this in the figure legends.
15. We have noted that the Western blot images have been spliced in panels Fig. 3B, Appendix Fig. S8A and S8D. Please mark all splices by using a visible line or space (e.g., a thin white or black line) to indicate where the blot was spliced. Please also provide uncropped source data images for these blots if available.
16. Papers published in The EMBO Journal are accompanied online by a 'Synopsis' to enhance discoverability of the manuscript. It consists of A) a short (1-2 sentences) summary of the findings and their significance, B) 3-4 bullet points highlighting key results and C) a synopsis image that is 550x300-600 pixels large (width x height, jpeg or png format). You can either show a model or key data in the synopsis image. Please note that the image size is rather small and that text needs to be readable at the final size.

With kind regards,

leva

leva Gailite, PhD
Senior Scientific Editor
The EMBO Journal
Meyerohofstrasse 1

D-69117 Heidelberg
Tel: +4962218891309
i.gailite@embojournal.org

We realize that it is difficult to revise to a specific deadline. In the interest of protecting the conceptual advance provided by the work, we recommend a revision within 3 months (31st Oct 2025). Please discuss the revision progress ahead of this time with the editor if you require more time to complete the revisions.

Referee #1:

My concerns have been addressed in the revised manuscript and I consider this interesting paper suitable for publication in the EMBO Journal.

Referee #2:

The authors have provided acceptable responses to my initial review comments and have appropriately altered the current manuscript. It is an interesting and valuable piece of work that should be well cited in the future.

Referee #3:

The authors have added additional data and have addressed the major concerns of the previous review. Specifically with regard to the structural data and supplementary enzymology experiments.

The authors addressed the remaining editorial issues.

Dear Jen,

Thank you very much for addressing the final editorial requests. I sincerely apologise for the delay in the processing of your manuscript due to the holiday period here in Germany and the resulting backlog. I am now pleased to inform you that your manuscript has been accepted for publication in the EMBO Journal.

Before we forward your manuscript to our publishers, we would like to propose some textual changes in the manuscript title, abstract and synopsis, mainly to improve its accessibility for our broader readership. I have also written a short blurb that will accompany the title of your manuscript in our online table of contents. Please take a look at the proposed edits in the attached text file and let me know if any corrections or adjustments are needed.

If you have any questions, please do not hesitate to contact the Editorial Office. Thank you for your contribution to The EMBO Journal, and congratulations on a nice study!

With best wishes,

Ieva
